# Generative Adversarial Neural Operators

**Md Ashiqur Rahman**                                                  *rahman79@purdue.edu*
*Department of Computer Science*
*Purdue University*

**Manuel A. Florez**                                                   *mflorezt@caltech.edu*
*Seismological Laboratory*
*California Institute of Technology*

**Anima Anandkumar**                                                  *anima@caltech.edu*
*Computing + Mathematical Sciences*
*California Institute of Technology*

**Zachary E. Ross**                                                    *zross@caltech.edu*
*Seismological Laboratory*
*California Institute of Technology*

**Kamyar Azizzadenesheli**                                            *kamyara@nvidia.com*
*NVIDIA Corporation*

**Reviewed on OpenReview:** *https://openreview.net/forum?id=X1VzbBU6xZ*

## Abstract

We propose the generative adversarial neural operator (GANO), a generative model paradigm for learning probabilities on infinite-dimensional function spaces. The natural sciences and engineering are known to have many types of data that are sampled from infinite-dimensional function spaces, where classical finite-dimensional deep generative adversarial networks (GANs) may not be directly applicable. GANO generalizes the GAN framework and allows for the sampling of functions by learning push-forward operator maps in infinite-dimensional spaces. GANO consists of two main components, a generator neural operator and a discriminator neural functional. The inputs to the generator are samples of functions from a user-specified probability measure, e.g., Gaussian random field (GRF), and the generator outputs are synthetic data functions. The input to the discriminator is either a real or synthetic data function. In this work, we instantiate GANO using the Wasserstein criterion and show how the Wasserstein loss can be computed in infinite-dimensional spaces. We empirically study GANO in controlled cases where both input and output functions are samples from GRFs and compare its performance to the finite-dimensional counterpart GAN. We empirically study the efficacy of GANO on real-world function data of volcanic activities and show its superior performance over GAN.

## 1 Introduction

Generative models are one of the most prominent paradigms in machine learning for analyzing unsupervised data. To date, there has been considerable success in developing deep generative models for finite-dimensional data (Goodfellow et al., 2014; Kingma & Welling, 2013; Dinh et al., 2014; Radford et al., 2015). Generative adversarial networks (GANs) are among the most successful generative models with rich theoretical and empirical developments (Arjovsky et al., 2017; Liu et al., 2017). The empirical success of GANs has been mainly within finite-dimensional data regimes; there has been relatively little progress on developing generative models for infinite-dimensional spaces–and importantly–function spaces. This is the case despite the fact that many fields of science and engineering, including seismology, computational fluid dynamics, aerodynamics,

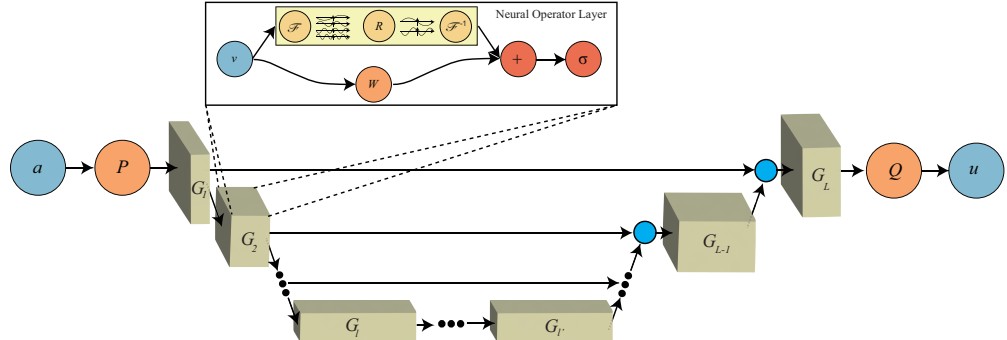

(a) Generator. $u = \mathcal{G}(a)$. The input $a$ first gets passed to a pointwise lifting operator parameterized with $P$. Then multiple layers of global integral operators $G_l$'s are applied which are accompanied by a few skip connections. At last, the output $u$ is generated using a final pointwise projection layer parameterized with $Q$.

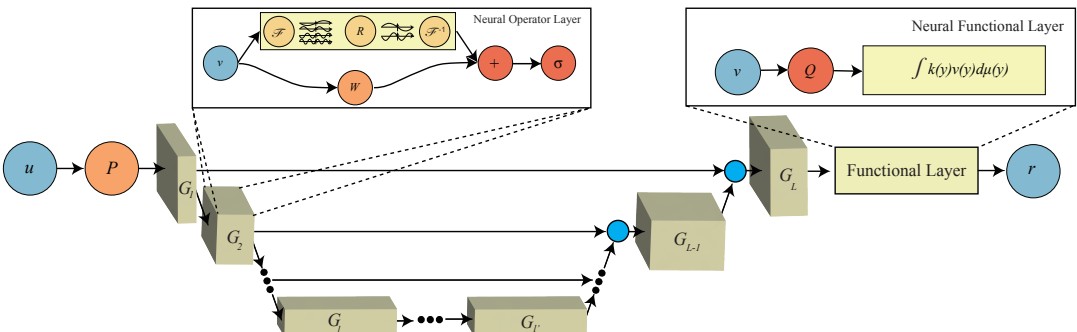

(b) Discriminator. $r = d(u)$. The input $u$ first gets passed to a pointwise lifting operator parameterized with $P$. Then multiple layers of global integral operators $G_l$'s are applied which are accompanied by a few skip connections. At last, the output $r$ is generated using a final pointwise projection layer parameterized with $Q$, followed by a linear integral functional layer.

Figure 1: Generative adversarial neural operator (GANO)

physics, and atmospheric sciences, work primarily with data that live in function spaces. In these settings, the observation data is mainly on irregular and changing points in both space and time.

Applications of functional data are abundant in seismology. For example, for a specific region on Earth, the base stations record data/seismograms on the surface of Earth. These receiver centers are located on irregular grids, e.g., point clouds (e.g., more stations closer to faults). The point cloud configuration also is different from region to region (Tokyo and Osaka). Moreover, due to measurement and local noise, some of the receivers are on and off in time. It means that we are dealing with functional data that are observed on irregular grids both in time and space. Moreover, when studying these functions, we aim to evaluate and query them at any spatiotemporal point. Since the governing equations are partial *differential* wave equations, access to the temporal and spatial derivatives reveals information about the dynamics of physical phenomena. Generative models for the mentioned wave functions allow for sampling many potential seismic behaviors of each region of Earth, facilitating the hazard study. Similarly, in weather forecasts, many recording stations on the surface of Earth are located on irregular grids (fewer stations on oceans than on lands) with different fidelity and frequency of observation. For these applications, scientists represent the weather condition as a function on a 2D sphere. This allows for evaluating weather conditions at any point on the surface of Earth and computing the gradient and momentum of the fluid dynamics. The function representation with a learned generative model allows for accurate sampling of weather forecasts and future events.

In this paper, we study the problem of generative models in function spaces. We propose generative adversarial neural operator (GANO), a deep learning-based approach that enables the learning of probabilities on function

spaces, and allows for efficient sampling from such learned models. GANOs generalize the GAN paradigm to function spaces, and in particular, to separable Polish and Banach spaces. GANO, unlike traditional kernel density estimation methods, is computationally tractable, works on general spaces, and does not require the existence of a density nor the assumption of defined underlying measures for density (Rosenblatt, 1956; Parzen, 1962; Craswell, 1965)[1]. Another line of work proposes to use neural stochastic differential equation (SDE) solver (Tzen & Raginsky, 2019) to generate temporal signal function with finite-dimensional co-domain (Kidger et al., 2021). However, while the generated signals are infinite dimensional objects, the loss construction in the mentioned work is still for finite-dimensional spaces, for grid evaluation points, therefore, making the learned generative model implicitly yet for finite-dimensional domains, and ergo, a special case of GAN setting. Such generative models require an underlying SDE solver to solve the temporal equation and are only designed for temporal data. The same SDE structure is also used for the discriminator models, resulting in causal discriminators, for which the optimality even for GAN setting is an open problem.

GANO consists of two main components, a generator neural operator and a discriminator neural functional. GANO architecture is empowered by neural operators, which are maps between function spaces (Li et al., 2020b). The generator neural operator receives a function sampled from a Gaussian random field (GRF) and outputs a function sample. This is in contrast to GAN, where the input is a sample from a finite-dimensional multivariate random variable and the output is a finite-dimensional object. The efficiency of traditional sampling methods from GRFs enables GANO to be considered as a computationally efficient generative model. The discriminator neural functional consists of a neural operator followed by an integral function. The discriminator receives either synthetic or real data as input and outputs a scalar. For the architecture choices in the generator, we use the efficient implementation of U-shaped neural operators (*U-NO*) (Rahman et al., 2022) and use Fourier integration layers, termed Fourier neural operator (FNO) (Li et al., 2020a) layers to construct push-forward maps from GRFs to the desired probability over function data. We use a similar architecture for the discriminator neural functional and use a three-layered neural network to implement the integral functional layer. For the adversarial min-max game, in particular, we instantiate the GANO framework by generalizing Wasserstein GAN (Arjovsky et al., 2017) setting to infinite dimension space. For the Wasserstein formulation, the discriminator neural functional is constrained to have a bounded norm in the infinite-dimensional space in terms of the Fréchet derivative operator. We propose how to impose this constraint in infinite dimensional space, which is invariant to the discretization. The discretization invariance property introduces one of the main differences to GAN setting where imposing the norm constraint requires hyperparameter tuning for each resolution and discretization.

The generator in GANO is a neural operator, a type of deep learning model that is resolution and discretization invariant Kovachki et al. (2021). It means that, the input function to the generator can be expressed with an arbitrary discretization or basis representation, yet the generated output is a function, which can be queried at any resolution or point. Similarly, the discriminator is a neural functional with input functions that can be expressed in any resolution or basis representation. These properties follow the recent advancements in operator learning that generalize neural networks that only operate on a fixed resolution (Li et al., 2020b; Kovachki et al., 2021).

The effective dimension of the output function space can be controlled by restricting the effective dimension of the GRF, e.g. by increasing the length scale of the defining covariance function. This is in contrast to GANs where the dimension of the input space controls the dimension of the output manifold. Table 1 compares the settings of GANOs and GANs. Since finite-dimensional spaces are special cases of infinite-dimensional spaces, and multi-variate Gaussian is a reduction of GRFs, then, GAN is a special case of GANO when applied on fixed grids.

We construct a series of controlled empirical study to assess the performance of GANO. To maintain full control of the data characteristics and complexity of the task at hand, we generate the data itself using GRFs of varying complexities. We show that GANO can learn probability measures on function spaces. One important example is when the data is generated from a mixture of GRFs; GANO reliably recovers the measure, while GAN collapses to a mode. In this work, we use the Wasserstein version of GAN for

---

[1]In finite dimensional spaces, it is conventional and standard to define density with respect to Lebesgue measures. However, in the infinite dimensional cases considered in this paper, Lebesgue measures do not exists and a density, if exists, needs to be defined with respect to a user-defined measure that the users need to argue for its relevance.

Table 1: GANOs and GANs

| Models | **GANO** | **GAN** |
|---|---|---|
| Input/output spaces | Function Spaces | Euclidean spaces |
| Input measure | Gaussian Random Fields | Multivariate random variables |
| Controls | length scale, variance, energy, etc. | dimension, variance, etc. |

compassion. We show that as the roughness/noisiness of the input GRF is increased, GANO properly learns to generate functions from the underlying data probability, while if the input GRF generates smooth or nearly fixed-value functions, the trained models lose the ability to properly capture the data measure.

We extend our empirical study to satellite remote sensing observations of an active volcano, where each data point is the phase of a complex-valued function defined on a 2D domain (Rosen et al., 2012). This is a real world function dataset in which each data point represents $\sim$ millimeter-scale changes in the surface of a volcano at a spatial resolution of $\sim$ 70 meters, measured every 12 days. This dataset constitutes a noisy and challenging function dataset for GANO and GAN training. We show that GANO learns to generate functions on par with the real dataset while GAN fails in generating these volcanic phase functions.

We release the code to generate the data sets in the first part of the empirical study. For the purpose of bench-marking, we also release the processed volcano dataset, which is ready to be deployed in future studies. We also release the implementation code along with the training procedure.

## 2 Related Works

The original GAN formulation can be interpreted as an adversarial game procedure in which the Jensen–Shannon divergence between a synthetic distribution, implicitly defined by a generator model, and a real data distribution is minimized (Goodfellow et al., 2014). However, models trained with a Jensen-Shannon objective function require substantial tuning, suffer from stability issues, and are notoriously difficult to scale (Radford et al., 2015). Considerable work has therefore been devoted to developing novel architectures, improving the formulation, and enhancing the theoretical understanding. In particular, the Wasserstein version of GAN allows for a more stable training scheme, is less sensitive to hyperparameter and architectural choices, and provides a loss function that correlates with output quality (Arjovsky et al., 2017). The Wasserstein formulation is often understood as an attempt to minimize the Wasserstein or Earth Mover's distance between the synthetic and real data distributions. In Adler & Lunz (2018), a rigorous theoretical extension of WGANs along with theoretically grounded choices of hyperparameters are presented, which the present paper follows. For the comparison study, we choose the Wasserstein version of GAN.

There has been limited previous work on learning densities over function spaces. These works have mainly focused on non-parametric density estimation with $\delta$-sequences on separable Banach spaces and topological groups (Rao, 2010; Craswell, 1965). Heuristic kernel density estimation for infinite-dimensional spaces was also developed (Dabo-Niang, 2004). Such methods assume the existence of a density with respect to (sometimes unspecified) base measures (Lebesgue measures are undefined for infinite-dimensional spaces) and impose strong assumptions on the metric and similarity of the output spaces. Moreover, learning the density does not provide matching algorithmic sampling methods from such infinite-dimensional spaces. Since pure memorization using $\delta$-sequences does not exploit the data structure and does not constitute a particularly appealing approach, we do not consider it an appropriate baseline for this study. For this study, we choose the GAN framework mainly due to its proximity to GANO, its vast success in many machine learning domains, and the lack of suitable methods for learning generative models in infinite dimensional spaces.

Pioneering work by (Li et al., 2020b) generalized the notion of neural networks to infinite-dimensional spaces and introduced the concept of neural operators, a novel composable architecture that is able to learn mappings between functions spaces. (Li et al., 2020a) showed that neural operators could be efficiently implemented as a series of convolutions performed in the Fourier domain of the input function. It has also been shown that any complex operator can be approximated by neural operators, which are compositions of linear integral operators and non-linear activation functions (Kovachki et al., 2021). Neural operators have been successfully

used for learning the solution spaces of Partial Differential Equations (PDE). FNOs have been used to learn the solutions to the Accustic Wave-equation in two spatial dimensions (Yang et al., 2021). Operator learning has transformed the field of physics-informed machine learning. (Li et al., 2021; 2020a) and improvements in the underlying architecture have allowed neural operators to learn complex solutions to multiphase flow problems (Wen et al., 2021).

A set of earlier attempts are made to develop learning methods to generate function samples using point cloud and point-wise evaluation of sampled data functions. Along these, neural process (Garnelo et al., 2018) is motivated as a Bayesian framework to generate function samples. While motivated as a generative model for underlying function distributions, the proposed amortized variational method aims at generating values of point sets, rather than functions. Therefore, it may come with a few limitations to be considered as a generative model for the underlying function distribution. This approach consists of an encoder model that given the point evaluation data, generates a finite-dimensional vector $z$ of noise which is aimed to be close to a prior multivariate Gaussian random variable. The noise $z$ is used as an input to an implicit neural network, i.e., the decoder. For any $z$, the implicit neural network represents a function sample that can be queried at any point on the domain. However, this early attempt does not learn the data distribution over functions and comes with a few limitations.

The proposed neural process method has limitation in the way the data is perceived, lacks expressively, and may not learn the underlying function distribution. As pointed out in the prior works (Dupont et al., 2021), the proposed neural process approach perceives the point cloud data as a set of values (no metric between points), therefore it ignores the presence of the metric space which is noted as a crucial limitation in prior works (Dupont et al., 2021). Furthermore, the proposed model maps a finite-dimensional $z$ vector to an infinite dimensional space of functions. Due to this limited input dimension, the generated functions can only cover a finite-dimensional manifold in function spaces, ergo, this method may lack the required approximation theoretic expressively to learn generating data functions. The last limitation is the fundamental issue with the formalism of the proposed neural process that prevents this method from being a generative model for the data function distribution. The proposed approach learns a model to maximize the probability of observing the values on the set of points rather than learning to generate function data. This is a major limitation of the proposed method and therefore, undesirable for the setting of learning function distribution. For example, consider a dataset consisting of many functions with a very low resolution (a few point evaluations) and only one function in the dataset with super high resolution (orders of magnitude more point evaluations, e.g., infinity). The objective of the neural process ignores the presence of all the function samples except the high-resolution one since it aims at maximizing the probability of point evaluation rather than the function samples. Moreover, since the formulation is Bayesian for points rather than functions, for a fixed number of function data samples, as we increase the number of point evaluations (e.g., to infinity), the prior will be ignored, and at the inference time, since the $z$ is drawn from the prior, even the generated points sample would not match the data. In the appendix, we show these limitations in a set of empirical tests, appendix A.1.

Another line of an attempt to learn generative models in function spaces proposes to accomplish the learning task in the space of *implicit neural network parameterize* of the given function space (Dupont et al., 2021). This approach proposes to train implicit neural networks to fit each data point in the data set. Ergo, for each data point, there will be a trained implicit neural network approximating it. Then a GAN model is trained to map input random vector $z$, e.g., drawn from a multi-variate Gaussian to the parameters of the implicit neural network. Ergo, for each draw of $z$, this approach computes the parameters of an implicit neural network, resulting in a function that can be queried at any point on the domain. This method requires extensive computation due to fitting many implicit neural networks and needs extensive memory to store these models. Furthermore, this model in the end is a map from finite-dimensional $z$ to infinite dimensional space, limiting its cover to the space of functions. In general, the proposed approach is a generic idea that has many favorable points as opposed to the prior works and does not have the fundamental and mathematical limitation of point samplings in prior works of neural processes (Garnelo et al., 2018). However, the current setting proposed in the prior work (Dupont et al., 2021) comes with a few limitations that prevent this approach to be considered as generative models of underlying function distribution.

The definition of the discriminator and the gradient penalty introduces fundamental mathematical limitations that prevent the model from learning the data distribution on function space. Given the construction of the discriminator, as the function resolution increases, e.g., the resolution goes to infinity, the proposed discriminator reduces to trivial maps, outputs a function instead of a number, and lacks the discrimination power desired for the learning task. The discriminator is implemented such that for an input function $u$, it computes $\sum_i W(x_i - x)u(x_i)$ where the summation is on the nearest neighbors and $W$ is a learnable multi layered neural network. As the resolution goes to infinity, i.e., on a uniform grid, $\sum_i W(x_i - x)u(x_i) \to W'(x)u(x)$. Therefore, repeating such pointwise operating layers many times results in a function with values at any $x$ equal to $\bar{W}(x)u(x)$, here $\bar{W}$ is the multiplication of $W'$s at all the layers. This pointwise architecture lacks expressive discriminating power. Moreover, the output of the discriminator is a function instead of a single number, which is undesirable since the discriminator is expected to output a number. The second issue is that, as the resolution of the function increases, e.g., goes to infinity, the gradient penalty merges to infinity and the training process misses learning the data distribution. A similar limitation is also observed in period works that use the stochastic differential equations (Kidger et al., 2021) to generate causal in-time data. This limitation makes the resulting models to be generative models for finite-dimensional spaces.

## 3 Generative Models in Function Spaces

One of the requirements to develop a stable model that maps an input probability measure to a general probability measure defined on infinite dimensional spaces is to have an infinite-dimensional input space. In this section, we describe the setting of such maps and propose GANO, a deep learning approach for learning generative models in infinite-dimensional function spaces. We propose GANO by extending the Wasserstein GAN formulation (Gulrajani et al., 2017) with a gradient penalty term applied to an infinite-dimensional setting.

### 3.1 GANO

Let $\mathcal{A}$ and $\mathcal{U}$ denote Polish function spaces, such that for any $a \in \mathcal{A}$, $a : D_{\mathcal{A}} \to \mathbb{R}^{d_{\mathcal{A}}}$, and for $u \in \mathcal{U}$, $u : D_{\mathcal{U}} \to \mathbb{R}^{d_{\mathcal{U}}}$. Let $\boldsymbol{G}$ denote a space of operators and for any operator $\mathcal{G} \in \boldsymbol{G}$, we have $\mathcal{G} : \mathcal{A} \to \mathcal{U}$, an operator map from $\mathcal{A}$ to $\mathcal{U}$. Let $\boldsymbol{L}$ denote a space of functionals such that for any functional $d \in \boldsymbol{L}$, we have $d : \mathcal{U} \to \mathbb{R}$, a functional map from $\mathcal{U}$ to $\mathbb{R}$.

Let $(\mathcal{A}, \sigma(\mathcal{A}), P_{\mathcal{A}})$ denote a probability space induced by a GRF on the function space $\mathcal{A}$. Following the construction of GRF, $P_{\mathcal{A}}$ is a probability measure such that for any sample $a \sim P_{\mathcal{A}}$ we have that for any finite collection of points $\{x_i\}_i$ in the domain $D_{\mathcal{A}}$, the joint probability of collection $\{a(x_i)\}_i$ follows a Gaussian probability. Furthermore, let $(\mathcal{U}, \sigma(\mathcal{U}), P_{\mathcal{U}})$ denote the probability space on the function spaces $\mathcal{U}$ that the real data is generated from. For a given function space $\mathcal{U}$, let $\mathcal{U}^*$ denote the dual space of $\mathcal{U}$. When $\mathcal{U}$ is also a Banach space, and $\mathcal{G}$ is Fréchet differentiable, we define $\partial \mathcal{G}$ as the Fréchet derivative of $\mathcal{G}$. For the measure $\mathbb{P}_{\mathcal{U}}$ and the pushforward measure of $\mathbb{P}_{\mathcal{A}}$ under map $\mathcal{G}$, i.e., $\mathcal{G}\sharp\mathbb{P}_{\mathcal{A}}$, we define the Wasserstein distance as follows,

$$W(\mathbb{P}_{\mathcal{U}}, \mathcal{G}\sharp\mathbb{P}_{\mathcal{A}}) = \sup_{d:d\in\boldsymbol{L}, Lip(d)\leq 1} \mathbb{E}_{\mathbb{P}_{\mathcal{U}}}[d] - \mathbb{E}_{\mathcal{G}\sharp\mathbb{P}_{\mathcal{A}}}[d] \tag{1}$$

For the dual space $\mathcal{U}^*$, we have that $Lip(d) \leq 1 \Leftrightarrow \|\partial d(u)\|_{\mathcal{U}^*} \leq 1$, $\forall u \in \mathcal{U}$. Therefore, we write the constraint in the form of an extra penalty part in the objective function, i.e.,

$$\inf_{\mathcal{G}\in\boldsymbol{G}} \sup_{d\in\boldsymbol{L}} \mathbb{E}_{\mathbb{P}_{\mathcal{U}}}[d(u)] - \mathbb{E}_{\mathcal{G}\sharp\mathbb{P}_{\mathcal{A}}}[d(u)] + \lambda\mathbb{E}_{\mathbb{P}'_{\mathcal{U}}}(\|\partial d\|_{\mathcal{U}^*} - 1)^2 \tag{2}$$

This relaxation is similar to the relaxation proposed in improved Wasserstein GAN Gulrajani et al. (2017) for finite dimensional spaces which recently have been shown to be equivalent to congestion transport (Milne & Nachman, 2022). In this objective, the constraint is induced as a soft penalty. The $\mathbb{P}'_{\mathcal{U}}$ is an uniform mixture of the data and generated data measures, i.e., $\mathbb{P}'_{\mathcal{U}} := \gamma\mathcal{G}\sharp\mathbb{P}_{\mathcal{A}} + (1-\gamma)\mathbb{P}_{\mathcal{U}}$ for $\gamma \sim U[0,1]$, where $U[0,1]$ is the uniform distribution in the interval $[0,1]$. Note that, while the cost functional in Eq. 2 is well defined, showing that the learned measure is indeed an approximation of $\mathbb{P}_{\mathcal{U}}$ remains an open problem. We address this issue empirically and perform a set of experiments that demonstrate that GANO produces diverse outputs from the data probability measure. To this end, algorithm. 14 summarizes the GANO training procedure. In

algorithm. 14, we first initialize the parameters of the generator $\theta_{\mathcal{G}}$ and the discriminator $\theta_d$. The learning process, at for each iteration, updates $\theta_{\mathcal{G}}$ and $\theta_d$ for $n_{\mathcal{G}}$ and $n_d$ times, respectively, each with $m$ samples to approximation the cost in Eq. 2.

## 3.2 GANO Architecture

We explain neural operators architecture as maps between function spaces. We describe the input and output of the generator which itself is a neural operator. We propose neural functionals that are maps from infinite dimensional spaces, e.g., function spaces, to finite-dimensional spaces, e.g., $\mathbb{R}$. Neural functionals are neural operators that are followed by kernel integral functional. We deploy neural functionals to implement discriminators.

**Neural operators** Neural Operators are deep learning models that are the building blocks of the generator and discriminator architectures in GANO to learn maps between function spaces, and the space of reals. Given an input function $a$ to a neural operator $\mathcal{G}$, we first apply a pointwise operator $\mathcal{P}$, parameterized with a neural network $P$, to compute $\nu_0$, i.e., $\nu_0(x) = P(a(x)) \ \forall x \in \mathcal{D}$. Let $\mathcal{D}_0$ denote the domain functions for which $\nu_0$ is defined in. Given the application of $\mathcal{P}$, we have $\mathcal{D}_0 = \mathcal{D}$. This point-wise operator layer is followed by $L$ integral layers. For any layer $i$, we have,

$$\nu_{i+1}(y) = \int_{\mathcal{D}_i} \kappa_i(x, y)\nu_i(x)d\mu_i(x) + W_i\nu_i(y) + b_i(y), \quad \forall y \in \mathcal{D}_{i+1}$$

where $\kappa_i$ is the kernel function, $d\mu_i$ is the measure in the $i$'th layer, $W_i$ is a pointwise operator, and $b$ is the bias function. This operation is followed by a pointwise non-linearity. The role of the pointwise operator $W_i$, aside from decomposing the linear operator to local and global terms, is similar to the residual connection in residual neural networks (He et al., 2016). We deploy convolution theorem to compute this integral as proposed in Fourier neural operator layer Li et al. (2020a). In particular, we write the $\kappa_i(x - y)$ and compute the first part of the integral operation in the Fourier domain. Let $\mathcal{F}$ denote the Fourier transform and $\mathcal{F}^{-1}$ the inverse Fourier transform operations. Given a periodic function $\mu_i$ (periodicity can be achieved by padding, a common practice in convolutional neural networks), the output of the layer is as follows,

$$\nu_{i+1} = \mathcal{F}^{-1}\left(R_i \cdot (\mathcal{F}\nu_i)\right) + W_i\nu_i + b_i$$

where $R$ is the Fourier transform of $\kappa$, and for each Fourier mode $k$, $R_i(k)$ is a matrix of learnable parameters. To improve computation complexity, after the Fourier transforms at each layer $i$, we keep Fourier modes up to $k_i^{\max}$. This allows for an efficient implementation of the layer and the presence of the residual connection $W_i$ makes sure all the Fourier components are passed to the next layer. This step, along with the residual connection $W_i$, allows the resulting Fourier neural operators to take into account all the Fourier components at each layer.

After $L$ above mentioned integral layers and computing $\nu_L$ defined on the domain $\mathcal{D}_L = \mathcal{D}$, we apply the final pointwise operator $\mathcal{Q}$, parameterized with a neural network $Q$. It is such that for any $x \in \mathcal{D}$, we have $u(x) = Q(\nu_L(x))$. When the input function is provided on a discretized domain, e.g., on a grid, we use the Riemannian approximation of the Fourier transform to compute the Fourier modes at each layer. When the input is provided on a regular and uniform grid, this operation can be accomplished using fast and memory-efficient methods such as fast Fourier transform, resulting in efficient implementation of the corresponding neural operators.

Neural operators output functions that can be queried at any point. Furthermore, they can be applied on input functions presented in many forms, e.g., presented as weighted sum of basis functions, or presented on a discrete set of points that includes regular and irregular grids. This property of neural operators is known as discretization invariance (Kovachki et al., 2021). In the following, we provide the definition of discretization invariance. Let $D_j$ denote a discretization (e.g., point cloud) of size $j$ in $\mathcal{D}_{\mathcal{A}}$. We call a sequence of nested sets $D_1 \subset D_2 \subset \cdots \subset \mathcal{D}_{\mathcal{A}}$ a discrete refinement of $\mathcal{D}_{\mathcal{A}}$ and each $D_j$ a discretization of $\mathcal{D}_{\mathcal{A}}$ if for any $\epsilon > 0$,

there exists a number $j \in \mathbb{N}$ such that,

$$D \subseteq \bigcup_{x \in D_j} \{y \in \mathcal{D}_\mathcal{A} : \|y - x\|_2 < \epsilon\}.$$

**Definition 3.1 (Discretization Insurance)** *For an operator $\mathcal{G} : \mathcal{A} \to \mathcal{U}$, where $\mathcal{A}$ is a set of m-valued functions, let $D_L \in \mathbb{R}^d$ be a L-point discretization of $\mathcal{D}_\mathcal{A}$, and for any $\theta \in \Theta$, a finite dimensional parameter space, $\hat{\mathcal{G}} : \mathbb{R}^{Ld} \times \mathbb{R}^{Lm} \times \Theta \to \mathcal{U}$ some map. We define the discretized uniform risk as,*

$$R(\mathcal{G}, \hat{\mathcal{G}}, D_L, \theta) = \sup_{a \in \mathcal{A}} \|\hat{\mathcal{G}}(D_L, a|_{D_L}) - \mathcal{G}(a)\|_\mathcal{U}.$$

*Given a discrete refinement $(D_j)_{j=1}^\infty$ of the domain $\mathcal{D}_\mathcal{A}$ we say $\mathcal{G}$ is discretization-invariant if there exists a sequence of maps $\hat{\mathcal{G}}_1, \hat{\mathcal{G}}_2, \ldots$ where $\hat{\mathcal{G}}_L : \mathbb{R}^{Ld} \times \mathbb{R}^{Lm} \times \Theta \to \mathcal{U}$ such that, for any $\theta \in \Theta$,*

$$\lim_{L \to \infty} R(\mathcal{G}(\cdot, \theta), \hat{\mathcal{G}}_L(\cdot, \cdot, \theta), D_L) = 0.$$

This definition implies that, as the discretization used to present the input function becomes finer, the approximate error in the approximate operator vanishes. It has been proven that neural operators are discretization invariant deep learning models and traditional neural networks fall short in this desirable property.

**Generator** We implement the generator operator $\mathcal{G}$ using an eight-layered neural operator model. The $\mathcal{P}$ point-wise operator consists of a one-layered neural network. The $\mathcal{Q}$ point-wise operator consists of a two-layer neural network. The parameter vector of the generator model is denoted by $\theta_\mathcal{G}$. The inputs to the $\mathcal{G}$ model are samples generated from a GRF defined on the 2D domain of $\mathcal{D} = [0, 1]^2$. The output of $\mathcal{G}$, and $u$'s are sample functions that are defined on a 2D domain. In this work, we utilize the *U-NO* architecture (Rahman et al., 2022) for its efficiency, stability, and robustness to the choice of hyperparameters. This architecture uses skip connections between layers and increases the dimensionality of the co-dimensions of the functions in the intermediate layers. Moreover, *U-NO* allows for highly parameterized models, a favorable property missing in the earlier Fourier neural operator models. In GANO, the generator neural operator model $\mathcal{G}$ outputs a function $u$ given a sample function $a$, i.e., $u = \mathcal{G}(a)$. Therefore, $\mathcal{G}$ pushes the GRF measure to a measure on the data space.

**Discriminator** The discriminator is a neural functional that consists of an eight-layer neural operator followed by an integral functional that maps the output function of the neural operator to a number in $\mathbb{R}$. In other words, we feed an input function $u \in \mathcal{U}$ to the neural operator part of the discriminator to compute the intermediate function $h$ and the output of the discriminator $r \in \mathbb{R}$ is computed as,

$$r := d(u) = \int \kappa_d(x)h(x)dx \tag{3}$$

where the function $\kappa_d$ is parameterized as a 3-layered fully connected neural network. Note that $h$ is the output of the inner neural operator with $u$ as an input, therefore, $h$ directly depends on $u$. The parameter vector of the discriminator model is denoted by $\theta_d$. The function $k_d$ constitutes the integral functional $\int \kappa_d(x)$ which acts point-wise on its input function. This linear integral functional as the last layer is the direct generalization of the last layer of discriminators in GAN models to map a function to a number. In many GAN models, the last layer maps a high dimensional vector to a number. This step is accomplished by a vector-vector inner product. Such a product, in continuum, resembles function-function inner product, i.e., the act of linear integral functional. This also directly follows the Riesz representation theorem Walter (1974) stating that, under suitable construction, a linear functional (map from infinite dimension to the space of reals) can be written as a linear integral functional. Fig. 1 demonstrate the architecture of the generator $\mathcal{G}$ and the discriminator $d$.

**Gradient penalty** In this paper, we consider the case where $\mathcal{D}_\mathcal{U}$ is a subset of a Euclidean space and to define the function space $\mathcal{U}$, we consider a measure $\mu$ on $\mathcal{D}_\mathcal{U}$. We often use Lebesgue measure for $\mu$ in this paper. The construction of the dual space of $\mathcal{U}$ and the computation of the Fréchet derivative used in the penalty term Eq. 2 follow after defining $\mu$. We represent the input function on a grid of $m_1 \times m_2$ (in general, it can be on any point cloud or basis function representation and following derivation follows). It allows us to use auto-grad to compute the gradient penalty for the Wasserstein loss. Following the function space definitions, the gradient penalty using the auto-grad call of $\nabla d(u)$ is implemented as $\mathbb{E}_{\mathbb{P}'_\mathcal{A}}(||\nabla d(u)||_{\{u(x_i)\}_i^{m_1 m_2}} - 1/\sqrt{m_1 m_2})^2$ which is different than the finite-dimensional view in GAN. The choice of $\sqrt{m_1 m_2}$ arises from the fact that we use the Lebesgue measure on $\mathcal{D}_\mathcal{U}$ to define the space $\mathcal{U}$. This ratio resembles that the basis functions deployed to represent the function $u$ on the grid of $m_1 \times m_2$ need to be chosen and scaled according to $m_1 m_2$ due to the fact that the basis functions are functions with unit norms in the metric space $\mathcal{U}$ with $L^2$ as the metric. It is important to note that since we compute the gradient with the consideration of the underlying metric space, the gradient computation using auto-grad is resolution invariant. Ergo, any resolution of choice can be used to train GANO models, fulfilling the premise of learning in infinite-dimensional spaces. Note that, for irregular grids where measures other than the deployed Lebesgue measures are used, this calculation should be adapted properly.

---

**Algorithm 1** GANO

1: **Input**: Gradient penalty weight $\lambda$, number of discriminator updates per iteration $n_d$, number of generator updates per iteration $n_\mathcal{G}$, number of samples per update $m$.
2: **Init**: Initialize generator parameters $\theta_\mathcal{G}$, discriminator parameters $\theta_d$, and optimizers $Opt_d, Opt_\mathcal{G}$
3: **for** each iteration $t = 1, \ldots$ **do**
4:      **for** $\tau = 1, \ldots, n_d$ **do**
5:          Sample $\{a_i\}_i^m$ from $\mathbb{P}_\mathcal{A}$, $\{u_i\}_i^m$ from $\mathbb{P}_\mathcal{U}$, and $\{\gamma_i\}_i^m$ from $U[0,1]$
6:          Compute loss $L := \frac{1}{m} \sum_i^m \left( d(u_i) - d(\mathcal{G}(a_i)) + \lambda(\|\partial d(u)|_{u = \lambda\mathcal{G}(a_i) + (1-\lambda)u_i}\|_{\mathcal{U}^*} - 1)^2 \right)$
7:          Update $\theta_d$ via a call to $Opt_d(L, \theta_d)$
8:      **end for**
9:      **for** $\tau = 1, \ldots, n_\mathcal{G}$ **do**
10:         Sample $\{a_i\}_i^m$ from $\mathbb{P}_\mathcal{A}$
11:         Compute loss $L := \frac{1}{m} \sum_i^m -d(\mathcal{G}(a_i))$
12:         Update $\theta_\mathcal{G}$ via a call to $Opt_\mathcal{G}(L, \theta_\mathcal{G})$
13:      **end for**
14: **end for**

---

## 4 Experiments

In this section, we study the performance of GANO when the data is generated from a GRF. We compare the performance of GANO against GAN in this setting. The models in GANO consist of eight-layer neural operators following the architecture in (Rahman et al., 2022). The initial lifting dimension, i.e., co-dimension is set to 16 and the number of modes is set to 20. To implement the GAN baseline model, we deploy convolutional neural networks, consisting of ten layers for the generator and half the size discriminator, and use Wasserstein loss for the training. For both models, we kept the number of parameters of the generative models roughly the same ($20M$). To train GAN models, we use GANO loss with the gradient penalty provided in the prior section. This choice is made to avoid otherwise required parameter turning for GAN loss for any resolution. We use the same grid representation of the input and output functions for the GAN and GANO studies. For training, we use Adam optimizer (Kingma & Ba, 2014) and choose a 2D domain of $[0,1]^2$ to be the domain where both input and output functions are defined on. For the empirical study, GAN is trained, optimized, and tested on a given discretization. Despite the fact that GANO can be trained and tested on any discretization, to make the comparison on par with GAN, we limit GANO experiment to the same discretization as GAN. It is worth noting that, GANO generates samples of functions that can be queried at any point and the GAN setting does not allow for it. Since GAN is not resolution invariant and does not generate function samples, it fails to be applicable to the general setting of function spaces.

We then study the effect of the roughness and smoothness of the input GRF on the quality of learning probability measures on function spaces for which we use the resolution of $\sim 64$ for each dimension. Lastly,

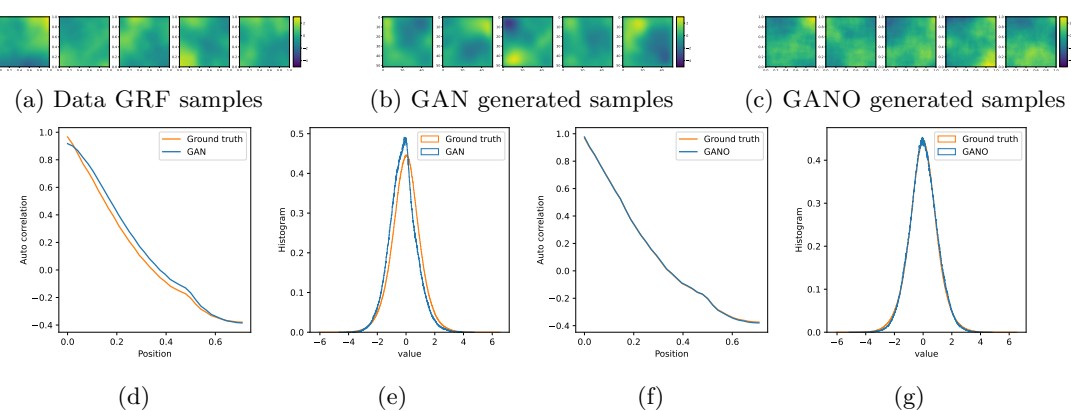

Figure 2: The input function sample is GRF and the data is generated from another GRF. (a) The samples of data GRF. (b) The samples of generated data from GAN model. (c) The samples of generated data from GANO model. (d) GAN Auto correlation. (e) GAN histogram. (f) GANO auto correlation. (g) GANO histogram

we study the performance of GANO on a real-world remote sensing dataset of an active volcano for which we use the resolution of ∼128 for each dimension. This is a challenging dataset with often times very low signal-to-noise ratio. For the choice of GRF, we choose the efficient implementation of Matérn based Gaussian process (Nelsen & Stuart, 2021) parameterized with $\tau$, the inverse length scale.

**GRF data**. For the setting where data is generated by sampling from a GRF, we use a dataset of random functions drawn from a GRF with length scale $\tau = 1$ (somewhat smooth functions). We use GAN and GANO approaches to learn the data GRF. We train the generative models using the inputs sampled from the same GRF Fig. 2. Fig. 2a demonstrate the sample data. Subsequently, Figs. 2b and 2c demonstrate the generated samples of GAN and GANO models respectively. To analyze the quality of the generated functions, we compare the auto-correlation and histogram of point-wise function values of the generated data and the true data, Fig 2. The $x-$axis in the histogram plots denote the values the functions take. The $x-$axis in the auto-correlation plots denote the positional distance of the points on the domain $\mathcal{D}_\mathcal{U}$ for which we compute the auto-correlation. We observe that GANO properly recovers the statistics of the data GRF in terms of auto-correlation, Fig. 2d, 2f, and the histogram of the generated function values, Figs. 2e, and 2g. We observe that, while the GAN approach provides smoother-looking functions, the functional statistics fail to be exact.

**Mixture of GRFs data**. For this experiment, we aim to learn to generate data from a mixture of GRFs. The training data is generated with an equal chance from either a GRF with a fixed mean function of 1 or −1, and both with $\tau = 1$. We use GAN and GANO approaches to learn the data probability measure, where the input functions are sampled from a mean zero GRF with $\tau = 1$, Fig. 3. Fig. 2a demonstrate the sample data. Subsequently, Figs. 2b and 2c demonstrate the generated samples of GAN and GANO models respectively. The auto-correlation and histogram of point-wise function values of generated data and the true data are provided in Figs, 2d, 2f 2e, and 2g. As we observe, GANO properly recovers the statistics of the data GRF in terms of functional statistics of auto-correlation and histogram. Similar to the previous experiment, we observe that the GAN approach provides smoother-looking functions, but in terms of the functional statistics, it drastically underperforms GANO.

In the previous two experiments, we observed that GANO enables us to learn measures on function spaces and generate samples that match the functional statistics of the underlying data. In the following, we examine the importance of the choice of input GRF on the performance of GANO.

**GANO and the length scale of the input GRF**. In GANO, when the GRF input to the generative model is very smooth (compared with the output GRF), we expect the generator to fail to learn a proper map. We expect this to be the case because the input lacks sufficient high-frequency components, and this smoothness

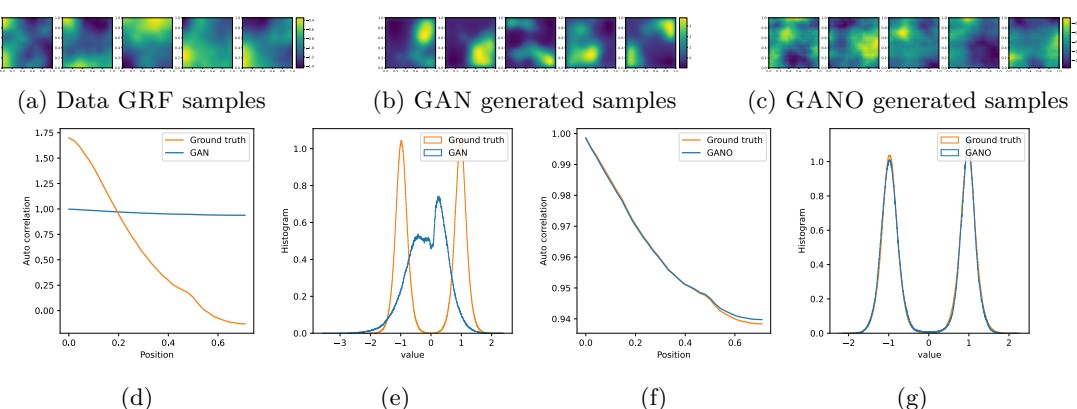

(a) Data GRF samples     (b) GAN generated samples     (c) GANO generated samples

(d)        (e)        (f)        (g)

Figure 3: The input function sample is GRF and the data is generated from a mixture of GRFs. (a) The samples of data from a mixture of GRFs. (b) The samples of generated data from GAN model. (c) The samples of generated data from GANO model. (d) GAN Auto correlation. (e) GAN histogram. (f) GANO Auto correlation. (g) GANO histogram

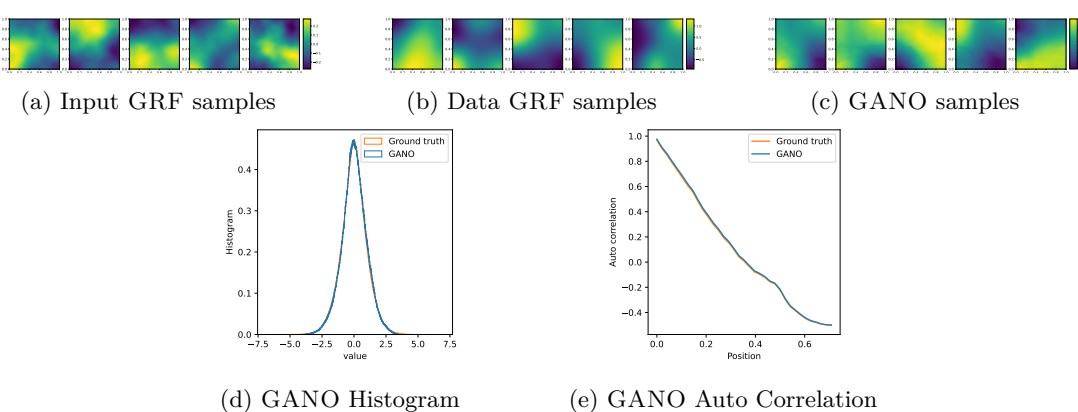

(a) Input GRF samples     (b) Data GRF samples     (c) GANO samples

(d) GANO Histogram        (e) GANO Auto Correlation

Figure 4: GANO trained on smooth data ($\tau = 5$) with rougher input GRF ($\tau = 7$)

prevents the generator from generating high-frequency and rough output functions. On the contrary, we expect that when the input GRF is much rougher than the data GRF and contains many high-frequency components, the generator would have an easier task to generate output functions. Therefore, the length scale and smoothness of the input GRF can play a role in regularizing GANO model, a very similar role that the dimension of the input multivariate Gaussian plays in the GAN approach.

We first show that when the input GRF is rougher ($\tau = 7$) and contains more high frequency components than the output GRF ($\tau = 5$), GANO successfully learns to generate samples with similar statistic of data GRF, Fig. 4.

When the output and input GRF are identical measures ($\tau = 5$), GANO still successfully learns to generate samples with similar statistics of the data GRF, Fig. 5. However, this setting requires more delicate hyper parameter tuning and requires more training epochs to converge. It is worth noting that, with proper choices of the spaces, an identity map may also be a solution.

Lastly, when the input GRF is smoother ($\tau = 3$) than the functions samples in the output data GRF ($\tau = 5$), the generative model fails to recover higher order statistics, including the auto correlation Fig.6. In this experiment the input function is much simpler than the output functions. This study suggest that, when the real function data is very complex, very noisy, contains varying high frequency components, and poses high

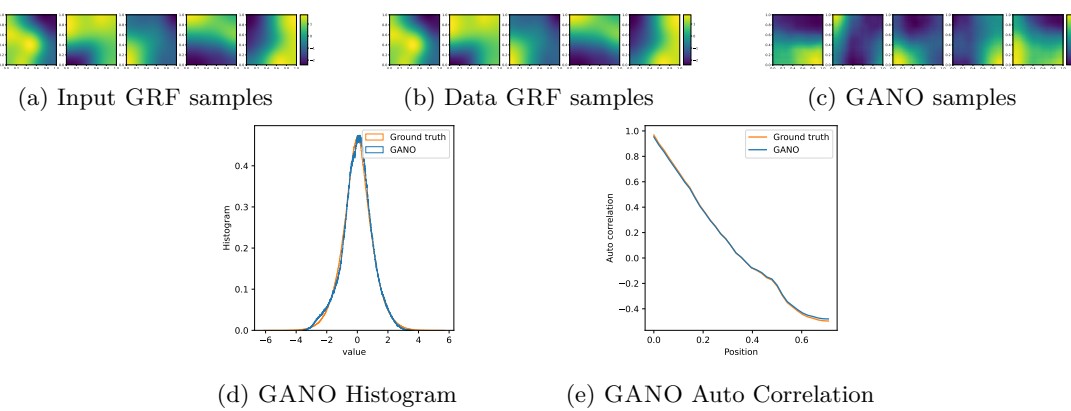

(a) Input GRF samples      (b) Data GRF samples      (c) GANO samples

(d) GANO Histogram      (e) GANO Auto Correlation

Figure 5: GANO trained on same GRF as input and data ($\tau = 5$)

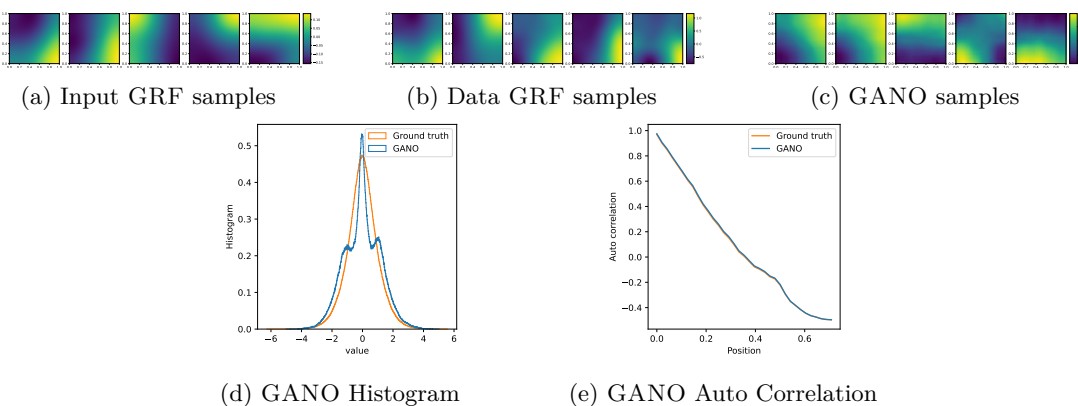

(a) Input GRF samples      (b) Data GRF samples      (c) GANO samples

(d) GANO Histogram      (e) GANO Auto Correlation

Figure 6: GANO trained on rougher data ($\tau = 5$) with smoother input GRF ($\tau = 3$)

entropy, it is crucial to provide the generator with on par GRF. On the contrary, when the function data at hand poses smoother behavior, a smooth GRF suffices for training a generator.

**Inputs and outputs of the generator in GANO are functions** The GANO framework is based on neural operators that are discretization invariant maps between functions spaces 3.1. The inputs to the generator neural operator model in GANO are functions and following the discretization invariance property of such models, these input functions can be provided to the generator in any discretization, and in particular in any mesh, grid, and resolution. In addition, the generator outputs functions, therefore, by definition, the outputs can be queried at any point in the domain. In the following empirical study, we demonstrate these properties of neural operators. We train the GANO models on one resolution and test the trained generator in another resolution. In particular, we consider a setting where for the training, the input GRF samples (with $\tau = 5$) are presented on a $64 \times 64$ grid on the two-dimensional domain. Moreover, the training data functions are draws from a GRF, with the same parameter as the input GRF, and samples are represented on the same $64 \times 64$ grid.

After training, we assess the above-mentioned properties of neural operators. We double the resolution of the input GRF samples and present them in a $128 \times 128$ grid. We provide these higher-resolution inputs to the generator to generate output functions. We evaluate the generated functions on a finer grid of $128 \times 128$. Figure 7 demonstrates the result of the study. Figure (c) represents high-resolution data, and figure (e) represents the generated samples on the higher-resolution input and query points. Figures (g) and (i) demonstrate the histogram and auto-correlation of the higher-resolution data and higher-resolution generated samples. This study expresses that neural operators can take inputs at any resolution and the

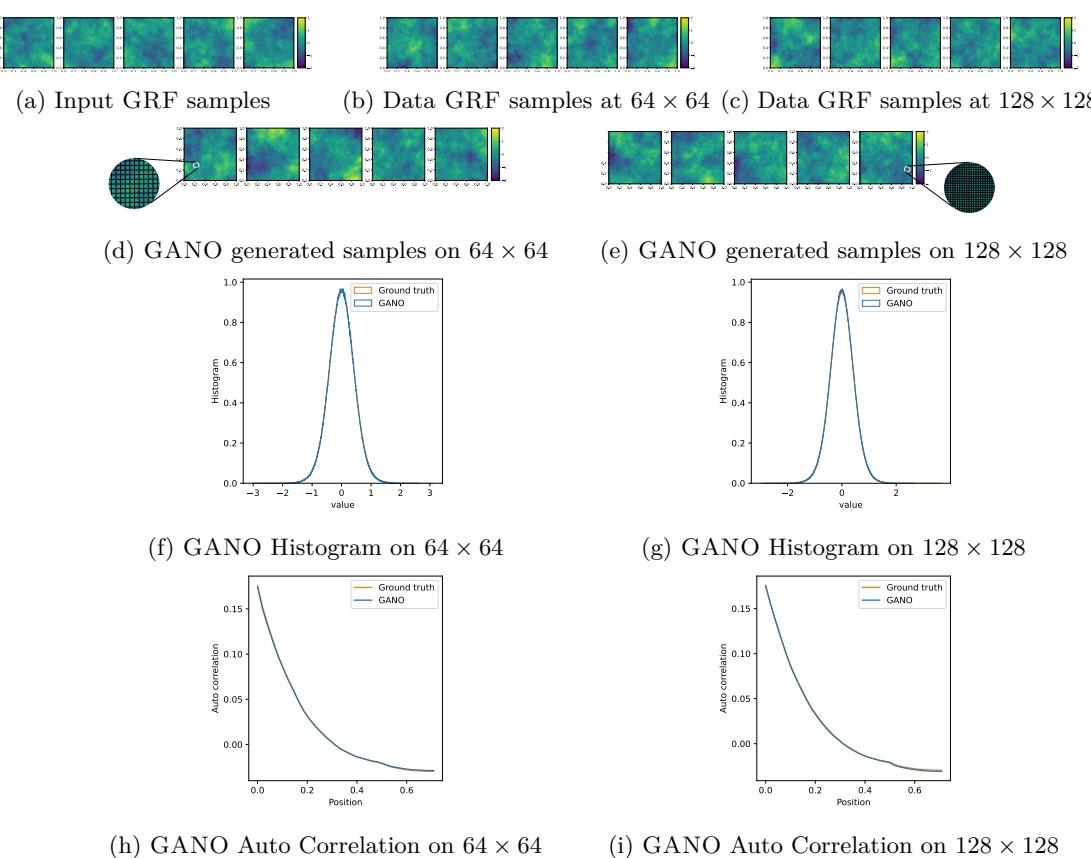

(a) Input GRF samples     (b) Data GRF samples at $64 \times 64$   (c) Data GRF samples at $128 \times 128$

(d) GANO generated samples on $64 \times 64$     (e) GANO generated samples on $128 \times 128$

(f) GANO Histogram on $64 \times 64$     (g) GANO Histogram on $128 \times 128$

(h) GANO Auto Correlation on $64 \times 64$     (i) GANO Auto Correlation on $128 \times 128$

Figure 7: We train GANO on a function data set of resolution $64 \times 64$. The data samples are generated using a GRF ($\tau = 5$). The input GRF ($\tau = 5$) sample functions are also represented on a $64 \times 64$ grid. The generator neural operator takes a function as an input and outputs a function. To demonstrate this fact, we test the trained generative model on a different resolution. We change the resolution of the input function to a higher resolution of $128 \times 128$ and query the generated function samples on a higher resolution of $128 \times 128$. Figure (c) represents high-resolution data, and figure (e) represents the generated samples on the higher-resolution input and query points. Figures (g) and (i) demonstrate the histogram and auto-correlation of the higher-resolution data and higher-resolution generated samples. This study expresses that neural operators can take inputs at any resolution and the output function can be queried at any point in the domain. Furthermore, despite the fact that the model has never seen high-resolution data during the training, it can generate statistically matching samples of high resolution.

output function can be queried at any point in the domain. Furthermore, despite the fact that the model has never seen high-resolution data during the training, it can generate statistically matching samples of high resolution. These are desirable properties of the GANO framework, as the first generative model on function spaces. Please note that, for this empirical study, we use smaller models in GANO in order to fit the high-resolution data to the present GPU machines. In particular, we reduce the number of layers to 5, the co-dimension to 8, and the number of modes to 10. These choices for the smaller model did not alter the performance of the trained generator.

**Volcano deformation signals in InSAR data.** Interferometric Synthetic Aperture Radar (InSAR) is a remote sensing technology used to measure deformation of Earth's surface, often in response to volcanic eruptions, earthquakes, or subsidence due to excessive groundwater extraction. In InSAR, a radar signal is emitted from satellites or various types of aircraft and echoes are recorded. Changes in these echoes over time (as measured by repeat flyovers) can be used to precisely measure the amount that a point on the surface moves between repeats. The most common form of InSAR data is the interferogram, which

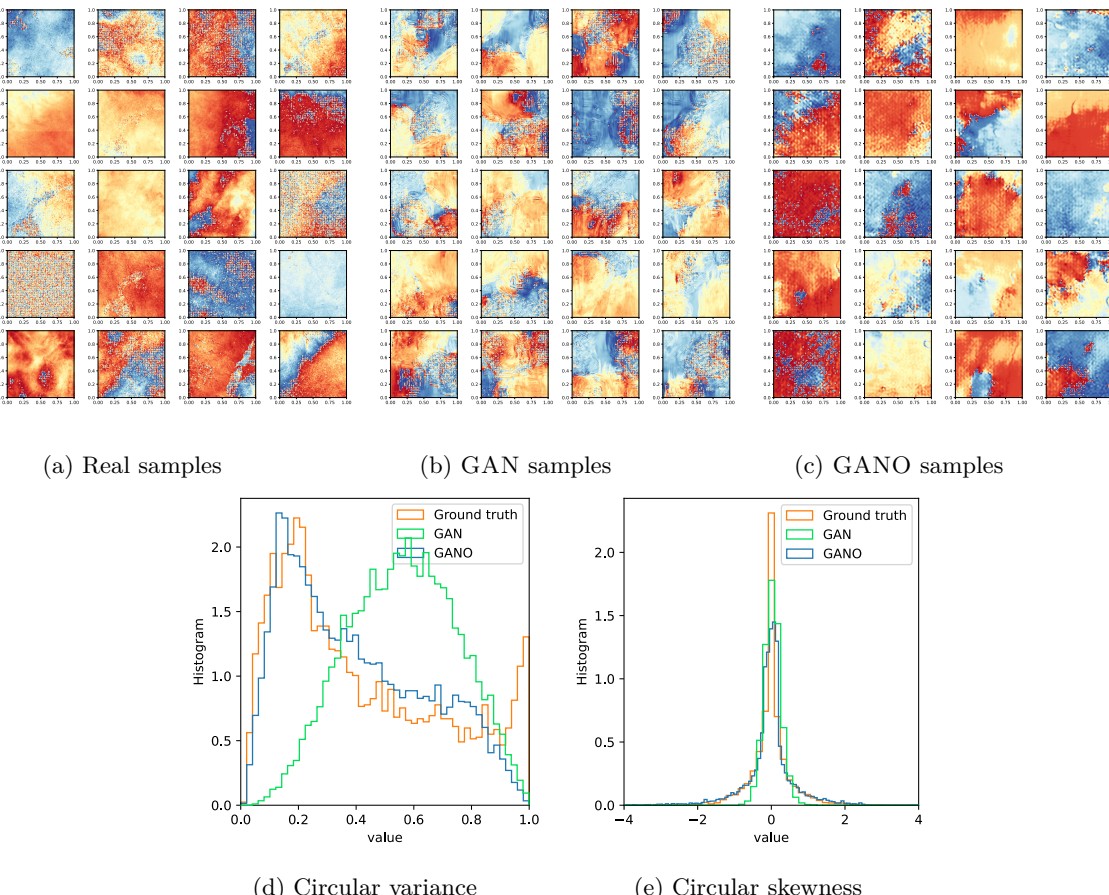

(a) Real samples          (b) GAN samples          (c) GANO samples

(d) Circular variance          (e) Circular skewness

Figure 8: GANO and GAN samples of InSAR data for Long Valley Caldera

is an angular-valued spatial field $u \in \mathcal{U}$, with $u(x) \in [-\pi, \pi]$ and $x \in \mathcal{D}$. Interferograms are known to be highly-complex functions because they exhibit many modalities, types of noises, and patterns that depend strongly on local atmospheric and topographic conditions. Additionally, since the values are angles on $[-\pi, \pi]$, if the change between two echoes is large enough, the angles can wrap around.

We produce a dataset of 4096 data points from raw interferograms, each in a grid of $128 \times 128$, from the Sentinel-1 satellites covering the Long Valley Caldera, which is an active volcano near Mammoth Lakes, California. We processed the InSAR functions/images, publicly provided by the European Space Agency, from 2014-Nov to 2022-Mar, covering an area around Long Valley Caldera (approximately 250 by 160 km wide) using the InSAR Scientific Computing Environment (Rosen et al., 2012). The stack of SAR functions is co-registered with pure geometry (precise orbits and digital elevation model) and the network-based enhanced spectral diversity approach. Then, we pair each function (277 in total) with its three nearest neighbors in time to form 783 initial interferograms with pixel spacing of 70 m. Finally, we subset each interferogram into six windows non-overlapping windows of $128 \times 128$ grid. Examples of real samples are shown in Fig. 8a.

We train GANO on the entire dataset of 4096 inteferograms. Generated samples are shown in Fig. 8c, where it is clear that many of the complexities of this dataset have been learned. One of the types of noise in interferograms results from decorrelation of the radar signal between repeat flyovers, and in the most extreme case, can lead to a stochastic process that is random uniform on $[-\pi, \pi]$ that covers part or all of the image. GANO is able to learn an effective operator that approximates this complex behavior. We quantitatively

evaluate the quality of the learned samples using circular statistics, which is necessary since these functions are angular-valued. Analogously to traditional random variables, there are moments of circular random variables. For a collection of $N$ random angular variables, $\{\theta_i\}_{i=1}^N$, define $z_p = \sum_j^N e^{ip\theta}$, where $i = \sqrt{-1}$. Then, $R_p = |z_p|/N$ and $\varphi_p = \arg(z_p)$. The circular variance is then given by $\sigma = 1 - R_1$, and the circular skewness is given by, $s = \frac{R_2 \sin(\varphi_2 - 2\varphi_1)}{(1-R_1)^{3/2}}$. Figs. 8d and 8e show the performance of GANO w.r.t circular variance and circular skewness. These results demonstrate the suitability of GANO framework on learning complex probabilities on function spaces and emphasizes the data efficiency of this framework.

For the comparison study, we train a GAN model on the same data set. Despite extensive hyperparameter tuning, the GAN model fails to learn to generate proper samples of functions. Fig. 8b demonstrates samples generated using a trained GAN model. The generated samples do not resemble the true samples, neither perceptually nor with respect to the circular variance and skewness Fig. 8d,8e. This study establishes the importance of learning the generative model directly in function spaces using global kernel integration instead of local kernels.

## 5 Conclusion

We propose GANO, a generative adversarial learning approach for learning probabilities on function spaces and generating samples of functions. GANO generalizes GAN, an established and powerful method for learning generative models on finite-dimensional samples. GANO framework consists of two models, a generator operator and a discriminator functional. We use the neural operator framework to directly model the generator and deploy the ideas from neural operators, and propose a new deep learning paradigm, namely neural functional, for the discriminator. We empirically show that the GANO framework is suitable for dealing with function spaces. We show that the input to the generative model can be chosen to be a GRF for which the length scale controls the diversity of the pushed measure. We release the code, package, datasets, and the results of this study for future reproducibility.

### Acknowledgments

The authors would like to thank the TMLR reviewers and the action editor for their constructive comments. Part of this research is developed when K. Azizzadenesheli was with the Purdue University. A. Anandkumar is supported in part by Bren endowed chair.

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

# A    Appendix

## A.1    Neural Process and generative functions

In this section, we expand the discussion on neural process approach (Garnelo et al., 2018) and show a few problems it may have in learning generative models for data function distribution. For a given function sample $u$, let $(x_i, y_i))_i^n$ denote its point evaluation representation, where for any $x_i$, a collocation point, $y_i$ is the point evaluation of the function at point $x_i$. For this construction, the following is the evidence lower-bound objective function proposed in neural process,

$$\log p\left(\{y_i\}_i | \{x_i\}_i\right) \geq \mathbb{E}_{q(z|\{x_i,y_i\}_i^n)} \left[ \sum_i^n \log p\left(y_i | z, x_i\right) + \log \frac{p(z)}{q(z|\{x_i,y_i\}_i^n)} \right]$$

where the method learns the encoder $q$ and a decoder map from $(z, x)$ to mean and variance of $p\left(y_i | z, x\right)$. This objective function maximize the probability $y_i$s, and does not give a formulation to learn the data function distribution.

Let's consider a trivial setting where the function distribution is a Dirac, meaning that, the data set consists of many repetitions of a single function. For example, consider the function $u(x) = 0.5$ on the interval

$[-\pi, +\pi]$. The dataset consists of many sample functions, all are the mentioned $u$. Following the Bayesian and variation form of this objective, when $n$ is small, e.g., $n = 100$ Fig. 9a, training this model results in a function distribution around the input function but does not collapse on the data function, Figure 9b.

For a reasonable generative model, we expect that, if we increase the function resolution, e.g., taking it to infinity, the function distribution learning approach to get better at learning a sensible generative model. However, in the heuristic neural process approach, as we increase the function resolution, the first term in the objective function dominates (goes to negative infinity), and $q(z|\{x_i, y_i\}_i^n)$ no longer can be replaced by $p(z)$ in the inference time. Figure 9c shows data function with 10000 point evaluations, and when neural process is trained on 10000 resolution input function, Figure 9d shows that the generated sample become more off and do not capture much about the function distribution.

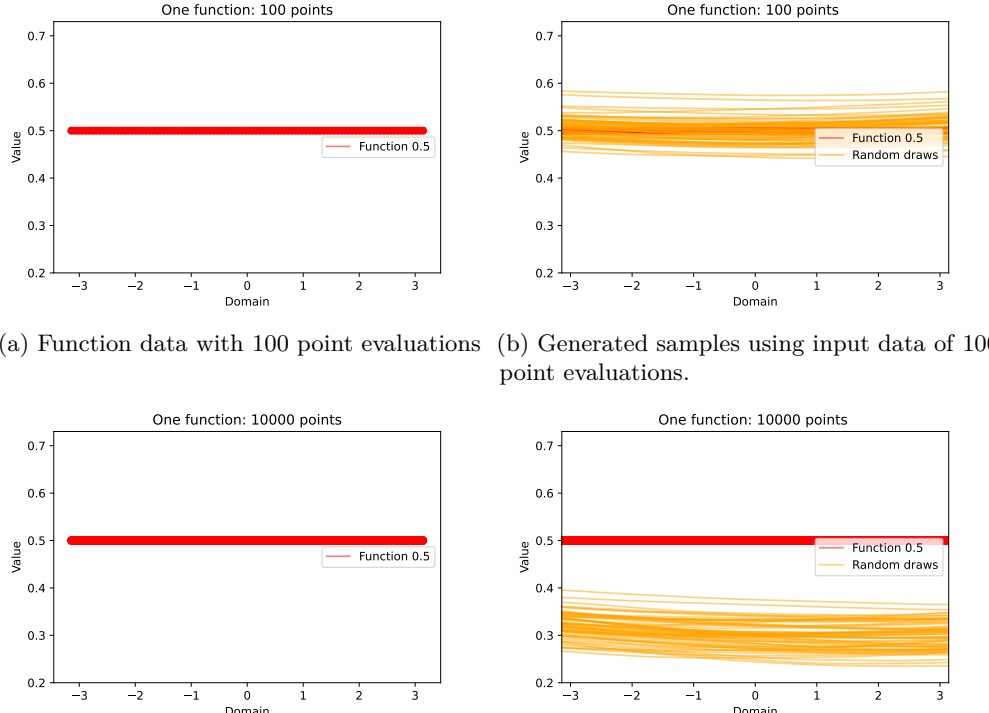

(a) Function data with 100 point evaluations   (b) Generated samples using input data of 100 point evaluations.

(c) Function data with 10000 point evaluations  (d) Generated samples using input data of 10000 point evaluations.

Figure 9: The sample generation in the neural process does not collapse to the data samples. As a Bayesian approach for point evaluations rather functions, as the number of point evaluations increases, the training process ignores the prior, resulting in an inconsistent model in the inference time.

To elaborate more on the inconsistency of the proposed heuristic neural processes model and its lack of foundations on leaning function data distribution, we construct the following additional toy example. Consider a similar setting as previous example with function distribution as a mixture of two Diracs on $u(x) = 0.5$ and $u(x) = 0.7$. In this setting, the data set consists of repetitions of $u(x) = 0.5$ and $u(x) = 0.7$ functions. A sensible function distribution learning method should be able to learn this mixture. Let us consider the setting where the resolution of $u(x) = 0.7$ is 2 (2 point evaluations), and the resolution of $u(x) = 0.5$ is 100. Per our above discussion on the lack of motivation behind the objection function proposed in neural process approach and the fact that this approach aims to capture point evaluation distribution, training on such mixture of data results in model that totally ignores the data samples of $u(x) = 0.7$. Figure 10a shows the data set and Figure 10b shows the learned model totally ignores the function samples $u(x) = 0.7$, only because they contain fewer point evaluations. We expect that, as the resolution of $u(x) = 0.5$ increases, neural process

approach even misses learning $u(x) = 0.5$. Figure 10c shows the data sets where $u(x) = 0.5$ has a resolution of 10000 and Figure 10d shows training on such data set does not learn the function distribution.

To this end, we concluded that the heuristic approach proposed in neural process does not learn function distribution, rather attempts to learn point evaluation, and it is not clear how this approach can be helpful to learn distribution of function data.[2]

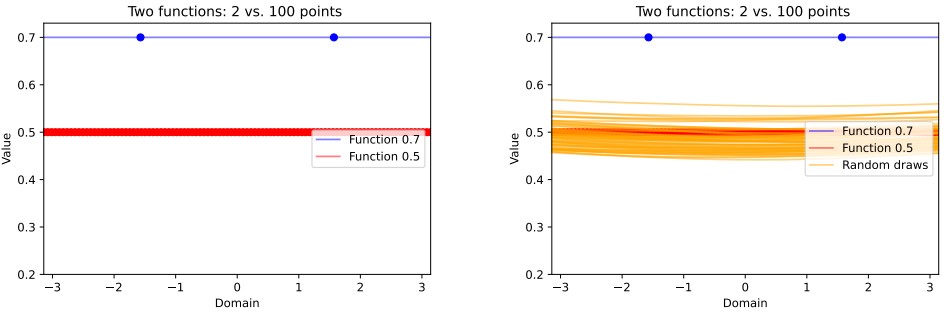

(a) Function data with 2 and 100 point evalua-  (b) Generated samples using input data of 2
tions                                            and 100 point evaluations.

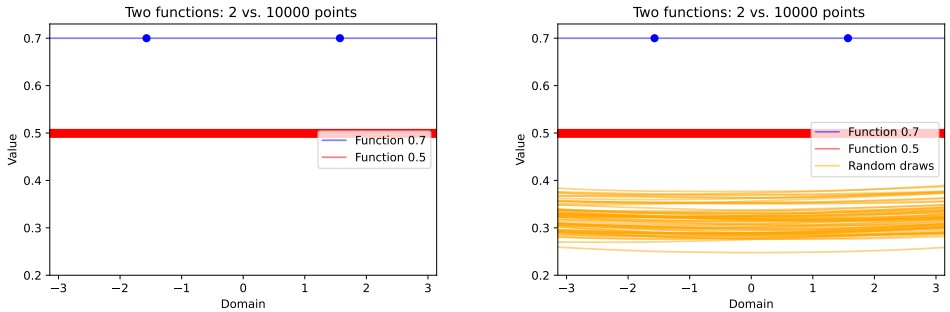

(c) Function data with 2 and 10000 point evalu-  (d) Generated samples using input data of 2
ations                                           and 10000 point evaluations.

Figure 10: Since the neural process approach aims to follow point evaluation distribution, the sample generated using the neural process ignores the low resolution data. Also, as the number of point evaluations in one function $u(x) = 0.5$ increases, the training process ignores the prior more, resulting in an inconsistent model in the inference time.

---

[2]For the empirical study on neural processes, we used the implementation provided in https://github.com/EmilienDupont/neural-processes

