# OpenReview forum: "Generative Adversarial Neural Operators"
_TMLR — Accepted by TMLR_

### Review · Reviewer_RQnM · 2022-08-20

**Summary Of Contributions:**

This paper proposes the GANO model, which is a generative model trained by a loss similar to WGAN on functional space. The model leverage previous neural operator model, and use it as the building block of the generator and discriminator. It is an interesting application of neural operator structure to generative modeling, and to my knowledge it is the first generative model on function spaces.

The paper studies GANO against GANs, the authors demonstrate the superior performance of GANO in modeling functions (with a discretization).


**Requested Changes:**

It would be good to address my concern in the weakness part, namely motivation for the approach, introduction to the details of neural operators, and more comprehensive choices of hyper-parameters.

**Strengths And Weaknesses:**

Strengths:

1. The problem of generative modeling for functional space is interesting and relatively less explored.

2. Strong empirical results of comparison with GAN.

3. Code release for potential reproduction.

Weakness:

1. Not sufficient motivation and application. While it is interesting to learn generative models in functional space, I think the task can be better motivated. For example, why is it important to train generative models for functions? What are some potential practical applications? What are the practical cases where traditional models on discretized grid does not work well?

2. Not sufficient background. Understanding the paper requires good familiarity with recent advances in deep learning in functional space, such as neural operators. To make the paper self contained, it would be better to briefly introduce the formulation of neural operators.

3. As pointed out in multiple previous research, to convincingly show the advantage over GAN, more choices of hyper-parameter setting need to be explored.

---

> ### Author Response · Authors · 2022-08-22
> **Motivation, neural operator architecture, and hpo is added.**
>
> We thank the reviewer for the constructive comments.
>
> 1) We extended the motivation of learning generative models for function spaces in the intro. Please refer to the second paragraph of the revised introduction. If there is any general application or suggestion that the reviewer has in mind that can help the paper, we would be grateful to the reviewer if they could share them with us.
>
> 2) Thank you for the suggestion. Now subsection 3.2 has a section on Neural Operators, followed by a section on Generator, Discriminator, and finally the gradient penalty.
>
> 3) We added a discussion on the comparison against GAN at the end of the first paragraph in section 4.
> We train, optimize, and test GAN on a given resolution.
> Despite the fact that \GANO can be trained and tested on any discretization, to make the comparison on par with GAN, we limited the \GANO experiment also to the same discretization as \GAN.
> Please note that \GANO generates samples of functions that can be queried at any point and GAN can't. Since \GAN is not resolution invariant and does not generate function samples, it fails to be applicable to the general setting of function spaces. We appreciate any comment on how to better convey this clarification in the paper.
>
>
>
> We appreciate the reviewer's comments and look forward to continuing this discussion
> Best,
> Authors.

---

### Review · Reviewer_rbkE · 2022-08-23

**Summary Of Contributions:**

This paper proposes a framework to extend the GAN framework [Goodfellow et al, 2014] to generate infinite dimensional objects such as functions using the architectures developed by [Li et al. 2020ab]. This work has application to modeling the distribution of some physical phenomena that are usually the solutions of some highly non-linear PDEs. The authors also show how to extend gradient penalty to function space with grid discretization and propose experiments on simulated and real physical data.


**Broader Impact Concerns:**

No concerns.

**Requested Changes:**

Answering the questions mentioned in the previous questions is critical to securing my recommendation for acceptance.

Mainly the following ones:
- The questions around the discretization and approximating integral by sums.
- The questions about the experiments.

thank you in advance.

**Strengths And Weaknesses:**

## Strengths

This work deals with an important problem of dealing with generating infinite-dimensional data, which can have important applications.

## Weaknesses

It seems that this work does not extend that easily (as we may think with a quick read) to non-uniform grids (see my questions).

The experiments could be more convincing. More especially, I think that providing a qualitative metric of performance would strengthen the results.

I feel that these two limitations should at least be transparently exposed in the paper.


## Questions:
- In equation (2) What is $\mathbb{P}’_{A}$ ? I guess it is a mixture between the data distribution and the generated distribution.
- It seems to me that there is some typos in the neural operator equation. First, it seems to me that $\mathcal D_i = \mathcal D$ (which is the case in [Li et al. 2020a] equation 2). Second, the integration should be with respect to $x$, i.e.,
$$ \nu_{i+1}(y) = \int_{\mathcal D} k_i(x,y)\nu_i(x) d \mu(x) + M \nu_i(y) + b_i(y)$$
Thus does it mean that we have to have $D_U = D_A$?
- I have an important question regarding approximating integrals by sums. Since you have only access to the evaluation of the real data samples $u_j$ at points $x_j$, i.e., $\{u_j(x_i)\}$. Thus, it means that when considering samples $u$ from $P_U$, you have to consider that the measure on $D_U$ is $\mu$ such that $x_j \sim \mu$ because the sum of the form $\sum_i f(x_i) u(x_i)$ only approximate $\int_{\mathcal{D}} f(x) u(x) d\mu(x)$. Thus, as mentioned, “Note that, for irregular grids where measures other than the deployed Lebesgue measures are used, this calculation should be adapted properly.” though I then think that implying that this work has application to irregular grids is a bit of an overclaim since, in my opinion, it is not clear that the framework extends and that the algorithm scales in that situation. I detail  my concern here:
  - In the case of irregular grid, it seems to me that the metric on the function space has to be the $L_2$ with respect to $\mu$ (the distribution of inputs) thus it mean that the target of the learning algorithm is to learn a distribution of functions that the discriminator cannot distinguish on that irregular grid. Thus, it seems very speculative to claim that “Similarly, in weather forecasts, many recording stations on the surface of Earth are located on irregular grids( fewer stations on oceans than on lands) with different fidelity and frequency of observation. [...] This allows for evaluating weather conditions at any point on the surface of Earth and computing the gradient and momentum of the fluid dynamics.” Overall I do not think that it is an argument for rejection since it does not seem that the previously addressed this concern, but I think it should be clarified in the paper's contribution that this work is on regular grid.
  - To the extent of my knowledge (and as mentioned in [Li et al,. 2020a] after (5) The FFT can only be computed when the grid is uniform) Thus, it is not clear whether the algorithm will scale with a non-uniform grid.
- In the experiments:
  - It is interesting that GANO fails at producing smooth functions. What is your interpretation of it? Do you think it is because you remove some modes in the FFT? Do you have experiments about the impact of this removal on the results (e.g. how did the tuning of $k_{\max}$ impact the results?
  - I am not 100% convinced by the experiments on real data:
  - Looking at the samples and the circular variance, GANO seems better
  - However, It is not clear that GANO is better looking at the statistics, More specifically for the Circular skewness.
  - The distribution of the circular skewness of the real data seems slightly different between Figure 7 and Figure 8, the max is a different value), is it because you used a different bin size? Thus I feel like the statement “The generated samples do not resemble the true samples, neither perceptually nor with respect to the circular variance and skewness” is an overstatement (more especially regarding the skewness).
  - To me, it seems like it lacks a more quantitative metric such as for instance a notion of distance between the generated distribution and the data distribution (this could be computed on the discretization which would end up being a distribution on “images”)




## Minor remarks:

“Note that, while the cost functional in Eq. 2 is well defined, showing that the learned measure is indeed an approximation of P U remains an open problem.” Note that https://arxiv.org/abs/2109.00528 shows that WGAN-GP computes a congested transport problem. (most likely not in the infinite-dimensional case though)

Some statements about GANs that I found inaccurate:
- “However, models trained with a JensenShannon objective function require substantial tuning, suffer from stability issues, and are notoriously difficult to scale”. Most of SOTA GANs use the original (non-saturating) formulation. For instance, [Broke et al. 2019], [Karras et al. 2021] [Sauer et al 2022]
- “One of the requirements to develop a stable model that maps an input probability measure to a general probability measure defined on infinite dimensional spaces is to have an infinite-dimensional input space.” For example, you could generate the parameter of a function thus pushing a measure from a finite-dimensional space to a measure on a space of functions.


## Typos:
“grids( fewer”

---

> ### Author Response · Authors · 2022-08-26
> **Measure to define the metric space and irregular grids**
>
> Thank you dear rbkE for the reviewer.
> We appreciate the point about the colt paper on congestion transport. We refer to this work in subsection 3.1.
>
>
> 1) \P'_A is in fact the mixture mentioned by the reviewer. We added its description at the end of subsection 3.1.
>
> 2) In [Li et al. 2020a], D=D_i=D_A=D_U are decided by choice and it is not neccery. In the neural operator model used in this current work, these domains are not the same in the intermediate layers. The reviewer is right, there was a typo in "x" and it is corrected.
>
> 3) We think there are three questions in this comment.
>
> 3-1 The first one is regarding the note on :
> "Note that, for irregular grids where measures other than the deployed Lebesgue measures are used, this calculation should be adapted properly."
> This statement is concerned with the subsection in computing the gradient penalty. Please note that the measure \mu is part of the definition of the function space U.  \mu is the measure on the domain which allows us to define the metric and norm on that function space.  It does not mean that the collocation points x_i's or the mesh are drawn from \mu. \mu basically is there to define the function space. Since \mu define the metric and the Frechet derivative is defined with respect to that metric space, then the Frechet derivative depends on the mu.
>
> Please note that, for any mesh and any measure, traditional methods in the numerical analysis allow us to compute an approximation of the Frechet derivative. In the paper, as an exercise, we show that if \mu is Lebesgue and the mesh is regular, then we can use the reweighted finite-dimensional gradient (e.g. by an autograd call) to compute the approximate Frechet derivative.
>
> If the measure is Lebesgue, but the mesh is not regular, different weightings should be used to go from autograd to Frechet derivative. Basically for a given mesh and given measure, there are different weightings to be computed and used to go from autograd to approximate Frechet derivative. Our point in "Note that, for irregular grids where measures other than the deployed Lebesgue measures are used, this calculation should be adapted properly." is about this fact. In this paper, we do not focus on enumerating these approximation methods.
>
>
> 3-2) This quote "Similarly, in weather forecasts, many recording stations on the surface of Earth are located on irregular grids( fewer stations on oceans than on lands) with different fidelity and frequency of observation. [...] This allows for evaluating weather conditions at any point on the surface of Earth and computing the gradient and momentum of the fluid dynamics." in the introduction is to motivate that function data is common in sciences. It is added at the request of the reviewer RQnM.
>
> 3-3. Li et al,. 2020a proposes to replace the integral kernel integration in the last equation of page 4, with integral  convolution operator. Li et al,. 2020a then evokes the convolution theorem to write this integral as multiplication in the Fourier domain. For this step, they need to compute the Fourier transform of the input function, i.e., for a given function "a", they need to compute int_D a(x) exp(-2\pi i omega x). If point evaluations of "a" is available (on any arbitrary mesh), this integral can be approximated using Riemannian-style approximation, i.e., \sum  a(x_j) exp(-2\pi i omega x_j)\Deltax_j. In Li et al,. 2020a, the authors state that, if {x_j}'s are on a regular grid, this summation can be computed using fast Fourier transform built-in packages. If the points are not on a regular grid, then we just compute that summation directly. Please note that the proposed method by Li et al,. 2020a is a neural operator, therefore, by definition, it is discretization invariant, both in input and output.

---

> > ### Comment · Reviewer_rbkE · 2022-08-31
> > **thank you for your answer**
> >
> > Thank you for your answer. I have some slight follow-up questions.
> >
> > 2. it seems from the equation
> > $$ \nu_{i+1}(y) = \int_{\mathcal D} k_i(x,y)\nu_i(x) d \mu(x) + M \nu_i(y) + b_i(y)$$
> > that $ \nu_{i+1}$ and $\nu_{i}$ have the same input domain. (except if M = 0). Can you develop 'D=D_i=D_A=D_U are decided by choice and it is not neccery. In the neural operator model used in this current work, these domains are not the same in the intermediate layers.' ?
> >
> > 3-3. '' then we just compute that summation directly.'' yes, but do we agree that it may quickly become intractable?

---

> > > ### Author Response · Authors · 2022-09-01
> > > **Clarifications**
> > >
> > > Thank you dear rbkE for the comments.
> > >
> > > 2. Mv_i(y) is derived by first applying the local operator M on v_i and then evaluating the resulting function at y.
> > > One example of changing the domain was used in https://arxiv.org/pdf/2204.11127.pdf where for a few layers, the input domain is [0,1] and the output domain is [0,0.5]. This is derived by writing Mv_i(y) = matrix(W) * (v(2y)) for any y\in[0,0.5], where the local operator M is parametrized with the matrix W (Please also refer to the footnote on page 5 of the mentioned paper).
> > >
> > > 3-3 We agree with the reviewer that this approach increases the computation complexity. We add this point about computation to the paper and refer to
> > > https://arxiv.org/pdf/2003.03485.pdf
> > > https://arxiv.org/abs/2006.09535.pdf
> > > that propose multiple ways to mitigate this computation complexity through e.g., subsampling or multi-grid approaches.
> > >
> > >
> > > Thank you for the comments.

---

### Review · Reviewer_cT3N · 2022-08-26

**Summary Of Contributions:**

The authors extend Neural Operators, NO, (Li et al. 2020) to the GAN framework. Instantiating both the generator and the discriminators as NOs. While GANs generators typically map from one vector to another, in the paper the authors learn to map from one function to another.

**Requested Changes:**

Please add a reference for this: “These properties follow the recent advancements in operator learning that generalize neural networks that only operate on a fixed resolution”

If “R is the Fourier transform if k. Then why is k an input to R in R_i(k)?

To clarify, is the gradient penalty computed on samples from the function?

[CRITICAL] It would be helpful to show the final losses used for training in an algorithm similar to Alg 1. in the WGAN and WGAN-GP paper.

[CRITICAL] Clarify the description of the discriminator implementation. Why is the integral function applied point-wise? Is r a logit?

Is the GAN baseline trained on samples from functions? It would be helpful to clarify this.

Results in Figure 2 are not convincing. There is only a small difference between (d) & (e) and (f) & (g). Does this hold over several run? I suggest that the authors repeat experiments several times to demonstrate that these effects are not the result of noise.

Figure 2(c) why do the GANO samples contain high frequency noise while the GAN samples do not? Is this an artefact of the Fourier transform?

[CRITICAL] GRF Data: It appears that both the input and target distributions are a set of random functions from a GRF is this is the case then the generator could just learn the identity? Especially given that they appear to be defined on the same 2D domain? Please give more details of the input and target distributions.

GRF data: please clarify the domain of the target function distribution.

Figure 2(a): Please clarify wether these are samples from the target (or input) distribution?

Typo: Pg 2. the lost construction

Please clarify if the baseline is GAN or WGAN. Since you use the WGAN loss it may be better to label baselines as WGAN.

[CRITICAL] GANO and the length scale of the input GRF: How are the input and target distributions defined? For Figure 4 demonstrate that GANO fails when the input distribution is more smooth that the target one, but performs well when the input distribution is more rough that the target distribution,  the authors should compare results with both smooth and rough input distributions, otherwise Figure 4 does not support this argument.

How does the experiments in Figure 5 differ for Figure 2? And if GANO’s input and target distributions are the same, then surely the model can just learn the identity?

Figure 6 is not referenced in the text. I believe is should be referenced in the last paragraph on page 8.

Figure 6: please also give details of the input and target distributions.

I would suggest combining Figure 6 and 4. I would suggest using the same target distribution and using different input distributions and showing GANO results on the same plot with each type of input distributions.

As a general comment, it would be helpful to see multiple runs to see the variance over seeds.

The authors do not provide key training details, what is the learning rate and the optimiser?

[CRITICAL] The authors claim that the model can be evaluated at different granularities and scales. The authors should demonstrate this on a toy task.

[CRITICAL] GANs should not be the only baseline in this section. I would suggest comparing to another suitable baseline. For example:
- Neural Processes (Garnelo et al.)
- Generative Models as Distributions of Functions, GASP (Dupont et al.)

Volcano deformation signals in InSAR data: The authors should combine GAN and GANO results on the same plot to make comparison easier. The authors should also compare to at least one other method that learns distributions over functions.

GANO allows one to sample functions from a target distribution, but its not clear how this is useful in a down-steam task; what does such a model enable?

Related Work:
The authors should also compare their works to Neural Processes (Garnelo et al), Generative Models as Distributions of Functions  (Dupont et al.), which claim similar benefits and show results on many more datasets.

**Strengths And Weaknesses:**

Strengths:
(1) It is interesting to see GAN's applied to learning distributions over functions.\
(2) Results in Figure 3 are a nice demonstration comparing GANs to GANO.\
(3) Authors show results on a complex, real world dataset.

Weaknesses:
(1) The implementation of the discriminator and gradient penalty is not clear - this could be improved by including an algorithm in the paper.\
(2) The experiments are very limited and the authors only compare to GANs (see more below).\
(3) They authors do not show any results on a down-steam task.\
(4) Many experimental details are missing.

---

> ### Author Response · Authors · 2022-09-08
> **The suggested clarifications are made in the paper.**
>
> Dear cT3N, thank you for the comments.
> Following your suggestion, we added the algorithm and more details on the empirical studies. We also would like to reiterate that the code to run the empirical study, along with all the experiment details provided there and the results are after multiple runs. We also incorporated the suggested clarification.
>
> We changed to notation for the kernel to kappa
> The gradient penalty is computed on the mixture of data and generated samples.
> The final loss for the setting where the grid is uniform and the measure used in U is Lebesgue and is provided in the algorithm.
> The integral function is the continuum version of the linear layers in the last layer of discriminators used in GAN.
> When the input and output both are the same GRF, the identity map is one of the solutions to the GANO min-max game in eq1.
> We added a statement that we use WGAN for the study in this paper.
> "GANO and the length scale of the input GRF" the experiment is run for fixed data roughness and making inputs either smoother or rougher. We added the experiment detail in the corresponding section.
> Figure 5 uses smoother functions than figure 2. Details are added.
> Prior works on neural operators, (Li et al.) established the fact that neural operators can be evaluated at any discretization. We also added the definition of the discretization-invariance to the paper.
> We added a discussion on Neural Processes (Garnelo et al.). The Neural Processes approach aims at learning the distribution of values of the point evaluations rather than data functions. This heuristic and Bayesian-inspired approach has three fundamental issues:
> 1) It is formulated to maximize the probability of observed point values rather than function samples.
> 2) As the number of evaluated points increases, e.g., goes to infinity, the prior will be ignored. This is due to the heuristic Bayesian formulation of the proposed method. Please note that as the resolution of a function goes to infinity (the number of evaluated points goes to infinity), the sample number is still one. The Neural Processes formulation does not capture this.
> 3) Due to the issue with the proposed formulation if we have "n" random samples of functions, and only one of them is presented in very high resolution, the neural processing approach ignores all the other data points and only focuses on the only high resolution (many more points) function sample.
> All of these points are undesirable for an approach to learning the distribution of function data.
> We elaborate on these points in the paper. We also run multiple experiments to further demonstrate the mentioned drawbacks. We first show that, when the data consists of only one function, the generative model does not collapse to this single (and constant) function. More importantly, the serious drawback of this method appears when we increase the resolution of the function samples. As we increase the resolution, the generated samples become less related to the data function.
> In the last experiment, we show that, when the data distribution is a mixture of two constant functions, one given in low resolution and one in high resolution, the neural process approach ignores the presence of low-resolution function samples. In general, these are undesirable behaviors of the neural processes and are caused by the lack of foundations in the proposed formulations.
>
> We added a discussion on the prior work of Dupont et al.. The proposed approach has two fundamental issues. 1) As the resolution increases (goes to infinity), the proposed discriminator f_out(x) =\approx \sum_{i\in neighbor} W(x-x_i)f_i becomes f_out(x) = D(f(x)) which is a pointwise differential operation applied on the input function f. This is a major limitation of the discriminator. The second major problem with the proposed method by Dupont et al. is the gradient penalty. As the resolution becomes higher, the gradient penalty keeps increasing (gradient at more points is computed) and as the resolution goes to infinity, only trivial functions satisfy the gradient norm constraint. These two limitations and drawbacks of this prior work. We elaborate on these points in this prior work.
>
>
> We would appreciate any other comment by the reviewer that would improve the quality of this paper.

---

> > ### Comment · Reviewer_cT3N · 2022-09-13
> > **Response to rebuttal.**
> >
> > 1. > The gradient penalty is computed on the mixture of data and generated samples.
> >
> > What is the "data" here? Are these samples from the functions in the training data? When you say generated samples? Are these the sampled functions or sampled data points from the sampled functions?
> >
> > 2. Thank you for including an algorithm.
> >
> > 3. The authors have not answered my question "Is r a logit?"?
> >
> > 4. The authors have not answered my question "Is the GAN baseline trained on samples from functions? It would be helpful to clarify this."
> >
> > 5. The authors have not addressed by comment that "Results in Figure 2 are not convincing"
> >
> > 6. The authors have not addressed the following critical concerns:
> > > [CRITICAL] GANO and the length scale of the input GRF: How are the input and target distributions defined? For Figure 4 demonstrate that GANO fails when the input distribution is more smooth that the target one, but performs well when the input distribution is more rough that the target distribution, the authors should compare results with both smooth and rough input distributions, otherwise Figure 4 does not support this argument.
> >
> > > [CRITICAL] The authors claim that the model can be evaluated at different granularities and scales. The authors should demonstrate this on a toy task.
> >
> > 7. The authors should ideally still compare to other methods that model distributions over functions.
> >
> >
> > Aside: In this rebuttal it was not clear which sentences were addressing which points in my original review and it would also help to reference more precisely any changes to the paper (e.g. reference Algorithm, Figure, or Section numbers).

---

> > > ### Author Response · Authors · 2022-09-19
> > > **Discussion**
> > >
> > > Thank you dear reviewer cT3N for the comments. We appreciate the time and effort you have dedicated to assessing the presentation and contribution of this work. Addressing your comments and suggestions very much improved the quality of the present paper. Please find the response to your comments in the following.
> > >
> > > “1. The gradient penalty is computed on the mixture of data and generated samples.
> > > What is the "data" here? Are these samples from the functions in the training data? When you say generated samples? Are these the sampled functions or sampled data points from the sampled functions?
> > > Answer: Yes, the reviewer is correct regarding data. Data refers to samples from the training data set of functions that are discretized. The generated samples are sampled functions from the generative model, which are also discretized. This description is also provided at the end of page 5.
> > >
> > > “2. Thank you for including an algorithm.”
> > > Answer: Thanks for the comment.
> > >
> > > “3- The authors have not answered my question "Is r a logit?"?”
> > > Answer: The discriminator outputs a number which is the output of the last linear layer (with no non-linearity). Please note that the discriminator is not a logistic model.
> > >
> > > “ 4. The authors have not answered my question "Is the GAN baseline trained on samples from functions? It would be helpful to clarify this."
> > > Answer: Yes, the input to GAN is grid evaluation (or discretization) of sample Gaussian random field (GRF). The grid evaluation of GRF is identical to multivariate Gaussian, which is a standard input to GAN models.
> > >
> > > “5. The authors have not addressed by comment that "Results in Figure 2 are not convincing"”
> > > Answer: Fig 4,5,6 show that if the input to the generator is smooth, then training the GANO model to capture rougher outputs is challenging. Therefore, the smoothness/roughness of the input GRF plays the role of effective dimension in the GANO setting and can be used to control the generative power of GANO. This is similar to the GAN setting where the generative power can be controlled by changing the dimension of the input multivariate Gaussian to the generator. Multiple runs are used to validate the observation which results in the same observation.
> > >
> > > “6. The authors have not addressed the following critical concerns: [CRITICAL] GANO and the length scale of the input GRF: How are the input and target distributions defined? Figure 4 demonstrates that GANO fails when the input distribution is more smooth than the target one but performs well when the input distribution is rougher than the target distribution, the authors should compare results with both smooth and rough input distributions, otherwise Figure 4 does not support this argument.”
> > > Answer: The reviewer is right. We did compare the results on both smooth and rough input distributions. (Figure 4: smooth GRF to data, Figure 6: rough GRF to data)
> > > In this section of the empirical study, the input and outputs are samples for GRFs. Each GRF is parametrized with the inverse length scale parameter \tau, where larger \tau means rougher sample functions and smaller \tau means smoother sample functions.
> > > In the three experimental studies presented in Figure 4,5,6, the data to be learned is generated by GRF with tau =5. Figure 4 shows the study where input GRF to the generator has \tah = 7 (rougher functions). Figure 5 shows the study where input GRF to the generator has \tah = 5 (same roughness as the data). Figure 6 shows the study where input GRF to the generator has \tah = 3 ( smoother than the data functions). This experiment shows that we can control the generative power of the GANO model by changing the roughness of the input functions. We added these details into the corresponding figure captions and also section 4.
> > >
> > > ------- to be continued

---

> > > > ### Author Response · Authors · 2022-09-19
> > > > **Discussion**
> > > >
> > > > ------- continued
> > > >
> > > >
> > > >
> > > >
> > > > “[CRITICAL] The authors claim that the model can be evaluated at different granularities and scales. The authors should demonstrate this on a toy task.”
> > > > Answer: Thank you for the comment. We have added a subsection to section 5, page 12, and Figure 7, with the title “Inputs and outputs of the generator in GANO are functions” where we discuss the reviewer’s comment and add new experiments.
> > > >
> > > > We train the GANO model on data of 64x64 resolution. We test the trained model on a grid of 128x128. This study demonstrates that the trained model can take inputs at a higher resolution and the generated function samples can be evaluated at a high resolution. We empirically observe that the generated samples at the higher resolution statistically match the high-resolution data.
> > > >
> > > > “7. The authors should ideally still compare to other methods that model distributions over functions.”
> > > >
> > > > Regarding the two mentioned works of Garnelo et al. and Dupont et al., please refer to the discussion in the related works (section 2) and the empirical study in appendix A.
> > > >
> > > > We show that the method proposed by Garnelo et al. does not learn the distribution of function data and instead attempts to learn the distribution of point cloud evaluations. We further show that this heuristic method even fails in learning the distribution of point cloud values.
> > > >
> > > > We also found that the method proposed by Dupont et al. has two major problems that prevent learning function data distribution. One of these issues further prevents us from running any experimental study on this approach.
> > > > 1) Ill-defined discriminator
> > > > 2) Resolution un-aware gradient penalty
> > > >
> > > > In the method proposed by Dupont et al., if we increase the resolution, the discriminator outputs multiple numbers instead of a single number. Furthermore, in the limit of the resolution to infinity, the discriminator outputs a function instead of a scaler. Since the discriminator does not output a number, this very issue prevents us from running any function-based empirical study. The second issue is in the way the gradient penalty is imposed in this work. As we increase the resolution, the gradient penalty shoots to infinity, unless the discriminator is a trivial function. Since the gradient penalty dominates the loss, the approach does not learn the function data distribution.
> > > >
> > > > We ideally would have liked to provide a comparison against methods that learn function data distributions. However, as also mentioned by RQnM, our paper is the first approach for generative models in function spaces. The only prior work on generative models in function spaces is based on pure memorization and uses a sequence of delta Diracs to present the function distribution. The mentioned pure memorization approach does not incorporate any learning beyond memorization and does not serve as a proper baseline for our work.
> > > >
> > > > Thank you again dear cT3N for your thoughtful comments and suggestions.

---

### Review · Reviewer_DpXW · 2022-09-03

**Summary Of Contributions:**

This submission proposes the GANO, as a function space generalization GANs. It places neural operators [1] in an adversarial framework and trains parametrised function-space mappings similar to GANs. Empirically, toy examples from Gaussian Random Fields and volcano deformation data (InSAR) are used to demonstrate the capacity of the new model, and its advantages over a GAN baseline.

[1] Li et al. 2018. Neural Operator: Graph Kernel Network for Partial Differential Equations

**Requested Changes:**

Address the weakness mentioned above.

**Strengths And Weaknesses:**

Strength

- Learning generative models in function spaces is interesting, and the combination of neural operators with GANs looks promising.
- The volcano deformation data demonstrates the potential practical impact of this work.

Weakness

The paper pitches GANO as a generalization of GANs, and claims its advantage over GANs. However, I think many of the claims are unfounded or based on unfair comparison.

- More serious baselines are needed to assess the proposed model.
    * The toy tasks are illustrative, but unfair for GANs as the data are generated from GRFs which were inputs into GANOs. As the GANO is proposed as a generalization of GANs, at the very least, similar toy tasks commonly used for GANs such as mixture of Gaussians, would help to level the playing field.
    * Given the existence of other function space models (such as [2]), additional baselines should be used to ground the performance of GANO in this domain. I am not convinced by argument in “Changes Since Last Submission” that GAN is the *only* meaningful baseline, given that it is by-design ill-suited for the tasks examined in the paper.

- The paper emphasizes at various places the advantage of function space modeling, and its accompanying “resolution and discretization invariance”. However, since any Fourier modes beyond the chosen max frequency are discarded, I think the model can be queried only at any *spatial* resolution, as the frequency resolution is limited. It is nice to provide this trade-off between spatial and frequency resolution, but I think the paper should be clear about this trade-off, instead of advertising it as an absolute advantage.

- A few other seemingly unfair comparison:
    * The paper claims “GANOs are more stable to train than GANs and require less hyperparameter optimization”. It is unclear to me as the effort of tuning GANO vs. the baseline GAN is not described in the paper.  On the other hand, it seems that GANO needs complex enough input GRF to match complex enough data (Fig. 4), while such input tuning is not required for GANs (which simply takes Gaussian noise in most cases).
    * The criticism of GANs: “The dimension of the input space controls the dimension of the output manifolds” is not true. See e.g., Appendix E of [2] for the robustness in the choice of input noise.

Further several experimental details are missing:
- I think the paper can benefit from a self-contained short introduction of the U-NO architecture on which the model based
- The implementation of gradient penalty on grids is unclear to me.
- How the architecture is chosen (e.g.What are the non-linearity and type of convolution)? In particular, contrary to the common practice of balanced generator and discriminator, why does the discriminator only have half the size of the generator?
- What is the “given discretization” used for all the experiments?

[2] Kidger et al. 2021. Neural SDEs as Infinite-Dimensional GANs
[3] Brock et al. 2019. Large Scale GAN Training for High Fidelity Natural Image Synthesis

---

> ### Author Response · Authors · 2022-09-08
> **Algorithm is added**
>
> Dear DpXW,
> We appreciate the comments. We added a descriptive algorithm to the paper along with the experiment details.
>
> 1-1 In this paper, we propose a method for generative models in function spaces. Our aim is not to show that the proposed method is better or worse than GAN in finite-dimensional spaces. Please note that the formulation in GANO reduces to the GAN setting when applied to finite-dimensional spaces ( Euclidean spaces are special cases of Banach spaces).
>
> 1-2. The proposed method in [2] is still a GAN model, learns distribution in finite-dimensional spaces, and is only proposed for casual time series.  In order to see the limitation of [2] in function spaces, consider a data set with many data points of low resolution, and one data point of very high resolution, e.g., infinite resolution. The gradient penalty in the mentioned work is infinity on that single infinite-resolution data point unless the discriminator is a trivial function. Therefore, the setting does not learn the proper generative model for data functions. This is due to the fact that the formulation is still based on finite-dimensional GAN. Please note that the proposed method in [2] is another GAN model applied to time series data.
>
> 2- Please note that this assessment "since any Fourier modes beyond the chosen max frequency are discarded" about FNO is incorrect. Higher frequency components are not discarded in the FNO model and are passed to higher layers using residual connections. Without these residual connections, the model had to only use the pre-specified Fourier basis functions, and wouldn't learn the input and output bases representations. This is not the case for generic neural operator models such as FNO.
>
> 3-1 Thank you for the great point. We took out this sentence from the abstract and elaborated on it in the intro. Basically, if we were to train a GAN model per resolution (since the framework is not resolution invariant),  a new hyperparameter is needed for the gradient penalty per resolution.
>
> 3-2 Lets consider a somewhat smooth function function f:R^d->R^d' for d<d'. The image of f in R^d' is at most d dimensional manifold. In order to control the dimension of the f-image manifold, one needs to tweak d. Similarly, in GAN, to control the dimension of the output manifold, one needs to tweak the dimension of the input. In the mentioned paper [3]., the authors show that for image datasets tried in that paper, d of 32 was sufficient.  Please let us know if we are missing your point here.
>
>
> Please let us know if any further comment comes up that you think would improve the presentation of this paper. We appreciate it.
>
> Best.

---

> > ### Comment · Reviewer_DpXW · 2022-09-18
> > **reply**
> >
> > Thank you for your reply.
> >
> > 1.
> > > Our aim is not to show that the proposed method is better or worse than GAN in finite-dimensional spaces.
> >
> > Thanks for reiterating this, but I think comparison with an appropriate baseline is still helpful for readers to understand the pros and cons of each method.
> >
> >
> > 2. Thank you for your clarification. I think it would therefore be helpful to more clearly describe these residual connections in the manuscript. (I tried search but cannot find any mentioning of "residual", and even "skip connections" only appears in the figure captions.) Besides, although
> > >Higher frequency components are not discarded in the FNO model and are passed to higher layers using residual connections
> >
> > What about the last layer?

---

> > > ### Author Response · Authors · 2022-09-19
> > > **Discussion**
> > >
> > > Thank you for the comments, dear reviewer DpXW. Your comments and suggestions helped us to improve the presentation and the quality of the present paper.
> > >
> > > 1) "Thanks for reiterating this, but I think comparison with an appropriate baseline is still helpful for readers to understand the pros and cons of each method."
> > > Answer:
> > > Thank you for the comment. We have now added experimental studies on methods aiming to learn distributions over function data (in appendix A) and also a discussion in the related work section (last three paragraphs).
> > >
> > > In particular, there are multiple papers that propose heuristic approaches to learning distributions of function data.
> > >
> > > The work of Garnelo et al., 2018 looks at the data as point clouds of values and attempts to learn the distribution of the values. This work does not learn the distribution of functions, and it even fails to learn the distribution of the point cloud values. We provide a detailed empirical study in the appendix.
> > >
> > > The work of Dupont et al., 2021 claims to learn a distribution over functions. However, this work comes with two major shortcomings. i) As one increases the resolution of the data functions, the discriminator no longer outputs a number which is a fundamental issue. As we tend the resolution to infinity, the discriminator outputs a function that is a pointwise local operation on the input function, implying a fundamental limitation of this heuristic approach. Because of this very limitation, we could not run similar function-based empirical studies that we investigated for Garnelo et al., 2018. ii) The second limitation is in the formulation of the gradient penalty. As we increase the resolution, the gradient penalty in the training loss goes to infinity, unless the discriminator is a trivial function. This is a similar issue shared with another approach proposed by Kidger et al., 2021.
> > >
> > > We incorporated these discussions into the paper showing that the mentioned works fall short in learning the distribution of function data. This makes the GANO framework the first work with a principled formulation that learns the distribution of function data.
> > >
> > > 2) We added that the pointwise operation W is on par with the residual connection in the ResNet models. Please refer to section 3.2.
> > >
> > > 3) "What about the last layer?"
> > > Answer: The last layer in the generator is the pointwise projection operator Q. Let's u:D->R^d denote the output function, and v:D->R^d' denote the input function to the last layer. For any x\in D, we take the d' dimensional vector v(x) and pass it through a two-layered neural network to compute a d dimensional vector. This d dimensional vector is u(x).
> > >
> > > Thank you again dear DpXW for your thoughtful comments and suggestions.

---

### Decision · Action_Editors · 2022-10-09

**Recommendation:** Accept with minor revision

**Comment:**

This paper introduces a new type of generative model for learning probabilities on infinite functional spaces. The method is novel and interesting. As stated in the introduction, there are real uses for this, and there is a convincing application of the proposed method to volcanic activities using real data. The paper has gone through a long list of improvements from reviews and the current version is significantly stronger than the first version I read.

Reviewers remain concerned of the lack of sufficient baselines, which in my opinion is a valid criticism especially given the narrative of the paper (e.g. constant comparison to GANs). The paper could benefit from another read through to reduce the tone on this front, given the lack of sufficient baselines of comparison. Nonetheless, as this is a new method, there is no truly proper baseline for comparison making it difficult to make said comparison. Reviews also remain concerned that it unclear how GANO can be used for a down-stream task or application.

Despite these outstanding criticisms, the paper is still worth publishing due to its novelty, convincing empirical study on real-world data, and fit to the journal audience.

Minor note that some mark-up needs to be removed for camera-ready copy (i.e. assuming the red color emphasis in the algorithm pseudo-code is not intended for the final version).

**Audience:**

Yes, this venue is a perfect fit for this paper.

**Claims And Evidence:**

Yes. The empirical study to volcanic activities is quite a convincing application of the technique proposed by the authors. The narrative is quite focused on the comparison to GAN, and as such, stronger baselines would be ideal. But since this is the first model of its kind, there are no perfect fit baselines to compare to.

That said, the paper does compare GANO's strengths over GAN often, inviting the criticism to the lack of sufficient baseline. The authors could make this implied by reducing the amount of comparisons to GAN where possible.